# Memristor-based biomimetic compound eye for real-time collision detection

Yan Wang[1,2,4], Yue Gong[1,4], Shenming Huang[1], Xuechao Xing[1], Ziyu Lv[1], Junjie Wang[1], Jia-Qin Yang[1], Guohua Zhang[1], Ye Zhou[3] & Su-Ting Han [1✉]

The lobula giant movement detector (LGMD) is the movement-sensitive, wide-field visual neuron positioned in the third visual neuropile of lobula. LGMD neuron can anticipate collision and trigger avoidance efficiently owing to the earlier occurring firing peak before collision. Vision chips inspired by the LGMD have been successfully implemented in very-large-scale-integration (VLSI) system. However, transistor-based chips and single devices to simulate LGMD neurons make them bulky, energy-inefficient and complicated. The devices with relatively compact structure and simple operation mode to mimic the escape response of LGMD neuron have not been realized yet. Here, the artificial LGMD visual neuron is implemented using light-mediated threshold switching memristor. The non-monotonic response to light flow field originated from the formation and break of Ag conductive filaments is analogue to the escape response of LGMD neuron. Furthermore, robot navigation with obstacle avoidance capability and biomimetic compound eyes with wide field-of-view (FoV) detection capability are demonstrated.

[1] Institute of Microscale optoelectronics and College of Optoelectronic Engineering, Shenzhen University, 518060 Shenzhen, P. R. China. [2] Hefei Innovation Research Institute, School of Microelectronics, Beihang University, 230013 Hefei, P. R. China. [3] Institute for Advanced Study, Shenzhen University, 518060 Shenzhen, P. R. China. [4] These authors contributed equally: Yan Wang, Yue Gong. ✉email: sutinghan@szu.edu.cn

Next generation of autonomous robots, especially equipped with artificial intelligence algorithm, will work in unrestricted dynamic environments from performing auto-drive in daily life to execute scientific expeditions at bottom of the ocean or outer space[1–5]. In this scenario, the key challenge for expanding the use of autonomous robots is perceiving and understanding their environment as well as adapting their behavior accordingly[6–11]. Artificial vision plays a vital role in informing robots of the impending collision[12–15]. For animals, there are three possible strategies for collision avoidance as: (i) to react when the approaching stimuli are away from it with a certain distance. This strategy requires the animals to approximate the depth based on the motion and binocular parallax. However, a lot of animals such as arthropods are improbable to utilize this approach due to their binocular fields as well as too small eye distance. (ii) To react at a certain time before collision by supervising the symmetrical expansion of the projected image without the exact knowing the distance $d$. As shown in Fig. 1b, there is an object subtending an angle $\theta$ at distance $d$ from the eye. The time before a collision can be calculated from the tau function. ($\tau_{(t)} = \frac{d}{d_{(t)}} = \frac{\sin\theta\cos\theta}{\theta'} \approx \frac{\theta}{\theta'}$ if $\theta$ is small) where $t$ is the time, $d$ is the distance, $\theta'$ is the angular velocity, and $\theta$ is the angular size. The tau function requires only knowledge of $\theta'$ and $\theta$. The firing rate of neuron can be decoded as $1/\tau_{(t)}$ which peaks at collision, as shown in Fig. 1d. In this strategy, an escape would be activated when $1/\tau_{(t)}$ exceeds a given threshold. Nevertheless, determining a threshold is difficult for biological systems. (iii) To track the approaching object by combining $\theta'$ and $\theta$ nonlinearly to induce a response profile, which follows the function $f(t)$ as shown in Fig. 1e. In the third possible strategy, the firing rate of neuron increases, peaks, and decreases when the collision happens to be imminent. Animals can anticipate collision in this way since the firing peak occurs before the image reaches its maximum size throughout the object approaching[16,17].

The lobula giant movement detector (LGMD), obeying the third strategy for collision avoidance, is a movement-sensitive, wide-field visual neuron placed in the third visual neuropile of lobula. LGMD can rapidly respond to looming object and efficiently triggers escape behavior. The response is typically more vigorous for fast-moving or small objects regardless of direction and is inhibited by large-field motions such as the flow field created by locust's own motion[18–22]. Therefore, locusts could move in crowded swarms containing millions of individuals with ultra-low-collision rates[23–25]. It is obvious that the principle derived from the LGMD might be ideal to design an artificial vision system of robot for collision anticipation through the implementation of neuromorphic hardware[26–28].

The vision chips inspired by the LGMD have been usually designed and implemented in very-large-scale-integration (VLSI) system which includes block of retinotopic units, static random access memory (SRAM) block, I/O registers and field-programmable gate array (FPGA) platform for addressing and timing control. More recently, a single device comprising monolayer $MoS_2$ photodetector stacked on top of floating-gate memory has been reported to emulate the behavior of the LGMD neuron[29]. The $MoS_2$ photodetector responding to the looming stimuli exhibits excitatory nature while the nonvolatile charge trapping capability of the floating-gate memory induces an inhibitive response. Integrating the excitatory response and inhibitory response in a single device has successfully mimicked

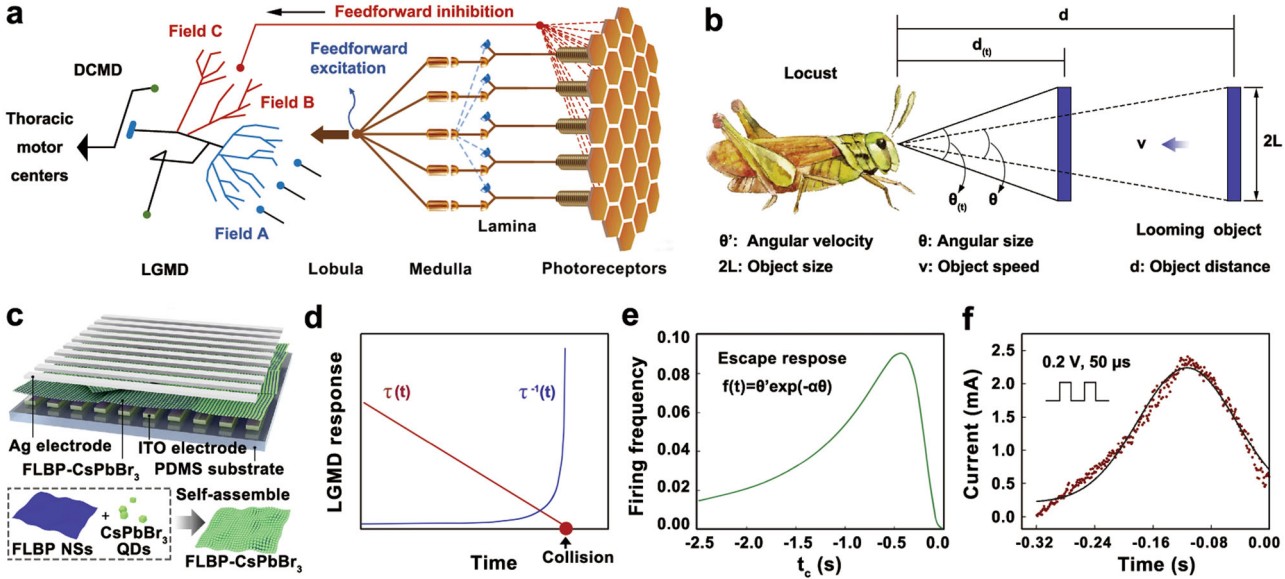

**Fig. 1 Biological and artificial lobula giant movement detector (LGMD) visual neuron. a** Schematic of the anatomical organization of the vision system with LGMD. The hexagonally packed photoreceptors of the locust's compound eye capture light stimuli and deliver the visual signal via the electrical impulses through the retinotropically arranged layers of lamina/medulla/lobula. Field A of LGMD receives the feedforward excitation, while field B and C receive the feedforward inhibition. Chemical synapse delivered from the LGMD neurons to descending contralateral movement detector (DCMD) to enforce reaching the thoracic motor centers to serve as a time tracking bio-system. **b** Motion-sensitive excitatory afferents to the LGMD neuron with subjecting visual stimulus. Looming subject with two high features of $\theta'$ and $\theta$ are conveyed through nonlinear mathematical operations $f(t)$ in LGMD in the form of dynamic firing frequency, which subsequently translates to anticipate collision. **c** Schematic diagram of the two-terminal threshold switching memristor (TSM) with the structure of Ag/few layer black phosphorous nanosheets (NSs)-CsPbBr$_3$ perovskite quantum dots (QDs) heterostructure (FLBP–CsPbBr$_3$)/indium tin oxide (ITO). CsPbBr$_3$ QDs were densely self-assembled in the FLBP NSs to form the FLBP–CsPbBr$_3$ heterostructured material. **d** Phenomenological model to analyze the evolution of looming objects in time of $\tau_{(t)}$ and $1/\tau_{(t)}$ to anticipate the collision time. **e** Response profile following the function $f(t)$ to track the approaching object with combined $\theta'$ and $\theta$. **f** Nonmonotonic response of artificial LGMD visual neuron implemented using TSM device with both voltage stimulus (0.2 V voltage pulse, 50 μs duration, 50 μs interval) and light stimulus (2.5 mW light power, 365 nm light wavelength).

the escape behavior of a LGMD neuron. The realization of mimicking LGMD neuron in a single device is more practical for robot's application compared with the VLSI system. However, transistor-based chips and single devices to simulate LGMD neurons make them bulky, energy-inefficient, and complicated. The devices with relatively compact structure and simple operation mode to mimic the escape response of LGMD neuron have not been realized yet.

In this work, the biomimetic compound eye composed of artificial LGMD visual neuron is implemented using $20 \times 20$ threshold switching memristor (TSM) arrays with a single device structure of Ag/few-layer black phosphorous nanosheets (NSs)-CsPbBr$_3$ perovskite quantum dots (QDs) heterostructure (FLBP–CsPbBr$_3$)/indium tin oxide (ITO) (Fig. 1c). Due to the optoelectronic coupling and charge transfer in FLBP–CsPbBr$_3$ heterostructure configuration, the threshold switching (TS) performance can be well controlled by the optical input. The biomimetic compound eye exhibits the wide field-of-view (FoV) detection capability ($180° \times 180°$) and nonmonotonic collision avoidance response to the looming stimuli. The excitatory and inhibitory response to the light flow field originated from the formation and rupture of Ag-conductive filaments (CFs) is analog to the escape response of LGMD neuron. To further present the application of artificial LGMD visual neuron that exploit these primitives, the robot navigation with obstacle avoidance is demonstrated.

## Results

**The algorithm for LGMD.** The morphology of LGMD neuron is shown in Fig. 1a. The biological LGMD neuron with large dendritic fanouts could receive feedforward excitation, lateral inhibition, and feedforward inhibition through the hexagonally packed photoreceptors of the compound eye, and retinotropically arranged layers of lamina, medulla, and lobula[30–32]. The large dendritic fan of LGMD (field A, Fig. 1a) receives the excitatory retinotopic input that could be further conveyed to the angular velocity $\theta'$ of the looming object. While the phasic non-retinotopic, feedforward inhibition correlated to the size of the object, $\theta$ is received by two dendritic fields arborize of LGMD neuron (field B and C, Fig. 1a). The phenomenological modes based on the angular velocity $\theta'$ and angular size $\theta$ measured at the retina have been widely investigated to describe the response of LGMD neuron to the looming objects by biologists[33,34]. The firing rate of LGMD ($f(t)$) in Fig. 1e can be fitted with the function of $\theta'$ and $\theta$ by multiplication of the inhibitory exponential of object size term and excitatory angular velocity term. $f(t) = \theta' \exp(-\alpha\theta)$, $\alpha = \tan^{-1}(2/\theta_{th})$, where $t$ denotes time, and $\theta_{th}$ is threshold angular size of the looming object at which the firing frequency of LGMD neuron achieves its peak value. When the object is far where $\theta$ is small, $\theta' \exp(-\alpha\theta)$ decreases slower than $\theta'$ increases, inducing the increase of $f(t)$. While the circumstance reverses due to the exponential dependence of $-\alpha\theta$ as the object approaches so that the activity of LGMD neuron peaks when the looming object achieves a certain angular size before the collision. In addition, LGMD is insensitive to translate objects since the motion-sensitive excitatory afferents to the LGMD are controlled by the presynaptic network-mediated lateral inhibition[31,35]. Above all, the LGMD neuron can anticipate collision efficiently owing to the earlier occurring firing peak before the collision and inhibited response to translate objects.

Figure 1f depicts the nonmonotonic response of our artificial LGMD visual neuron (Ag/FLBP–CsPbBr$_3$/ITO-structured TSM) that allows the mediation of both voltage and light. Output current exhibits a time-resolved excitation and inhibition behavior, which is analog to the general escape response toward

the approaching object in Fig. 1e. LGMD-like behavior of the device provides an effective and alternative approach to robot navigation with obstacle avoidance, which will be elucidated in the following sections.

**FLBP/CsPbBr$_3$ TSM.** The vision systems based on the simple algorithm of LGMD neuron for autonomous robot systems have been widely reported. Transistor-based chips and single devices attempt to simulate the $f(t)$ function representing the collision response of LGMD neurons, making them bulky, energy-inefficient, and complicated. To build LGMD-inspired artificial neuron, we used a single two-terminal volatile memristive switch with a stack structure of 30 nm Ag/18.6 nm FLBP–CsPbBr$_3$ heterostructure/100 nm ITO. The FLBP NSs and CsPbBr$_3$ QDs were synthesized according to the previous reports[36,37]. The self-assembly of as-synthesized CsPbBr$_3$ QDs onto the FLBP NSs was accomplished by bath-sonicating the mixed solution of CsPbBr$_3$ QDs and FLBP NSs at room temperature (see "Methods"). Transmission electron microscope (TEM) images in Supplementary Fig. 1 reveal that the CsPbBr$_3$ QDs are densely assembled on the FLBP NSs. Supplementary Fig. 2 shows the cross-sectional scanning electron microscopy (SEM) image of FLBP–CsPbBr$_3$ TSM in which the continuous and smooth surface of each layer was observed. Figure 2a shows the typical $I–V$ characteristics of FLBP–CsPbBr$_3$ TSM under the voltage sweep from 0 to $+3$ V/$-3$ V with the compliance current ($Icc$) of $10^{-3}$ A during the measurements. The FLBP–CsPbBr$_3$ TSM initially exhibited a resistance of $1.6 \times 10^9 \, \Omega$ at a high-resistance state (HRS) with a reading bias of 0.1 V (Fig. 2a). As the voltage increased to 1.05 V, the memristor was switched to a low-resistance state (LRS) with the on-current reaching the $Icc$ level. An extremely large switching ON/OFF ratio of over $10^7$ was observed. After removal of the applied voltage, the memristor returned back to its HRS spontaneously with a sudden current drop at 0.20 V. The response of the FLBP/CsPbBr$_3$ TSM to the direct current (DC) voltage sweeps displays the typical TS behavior. Almost symmetric hysteresis loops were observed in both positive and negative scans, suggesting the unipolar TS characteristic which is significantly different from nonvolatile electrochemical metallization memory (ECM). The active metal electrode material of Ag may be doped into the FLBP–CsPbBr$_3$ layer during the voltage sweeping process[38]. As the filament formation in the memristor is a "threshold" event governed by the net Ag numbers in a given filamentary volume. When the Ag CFs voluntarily rupture, the large amount of Ag atoms remains near the ITO electrode side. When the polarity of bias is reversed, they can act as a new Ag electrode to accomplish a reverse TS behavior and the bidirectional TS can be realized in our TSM device. Narrow distributions of the threshold voltage ($V_{th}$, $1.20 \pm 0.22$ V ($\pm$ s.d.) and $1.16 \pm 0.24$ V ($\pm$ s.d.) for the positive and negative polarity, respectively) together with Gaussian fits to the histograms are depicted in Fig. 2b. Accordingly, the variabilities of $V_{th}$, described as $\sigma/\mu$ ($\mu$ is the mean of the $V_{th}$ distribution and $\sigma$ is the standard deviation), were calculated to be 0.234 and 0.150 for positive and negative voltage sweeping, respectively. Independent relationship was highlighted between the $V_{th}$ and $Icc$ in Fig. 2c and Supplementary Fig. 3, implying a field-driven mechanism of the FLBP–CsPbBr$_3$ TSM.

The dynamic properties of FLBP–CsPbBr$_3$ TSM were investigated in Fig. 2d by applying the voltage pulses and recording the resulting current[39–43]. Following the applied voltage pulses of 1 V, the current jumped suddenly to $Icc$ of $1 \times 10^{-5}$ A. The drift and discharge of Ag$^+$, as well as subsequent growth of Ag CFs, leads to the transition from HRS and LRS. As the voltage pulse ended, the TSM decayed to its original LRS over a relaxation time of

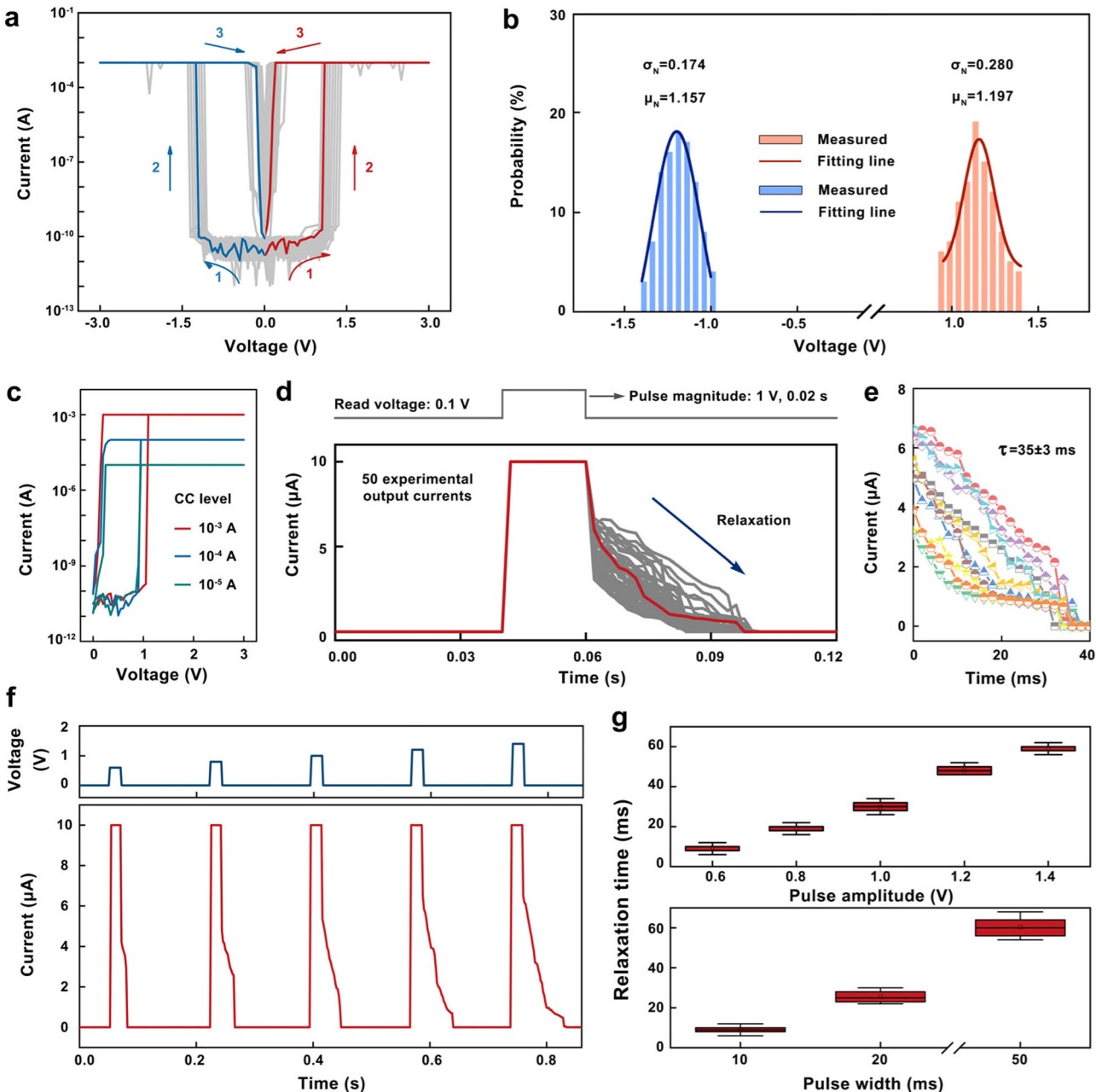

**Fig. 2 Threshold switching (TS) behavior in FLBP–CsPbBr₃ TSM. a** Unipolar TS characteristics of 100 TSM cells. **b** Histogram of $V_{th}$ of FLBP–CsPbBr₃ TSM under positive sweep (red) and negative sweep (blue) with Gaussian fitting. **c** I–V characteristics of FLBP–CsPbBr₃ TSM under positive sweep with $10^{-3}$, $10^{-4}$, $10^{-5}$ A compliance current (Icc) level. **d** Repeatable relaxation features in the FLBP–CsPbBr₃ TSM for pulsed (1 V, 20 ms) tests. **e** Current decay process resulted after each input action in (**d**) with fitted decay time constant $\tau$ from the stretched-exponential function. **f** Electric pulses induced switching behavior in the FLBP–CsPbBr₃ TSM. **g** Distribution of delay time for input pulse with different pulse amplitude (0.6–1.4 V with 20 ms pulse width) and pulse width (10–50 ms with 1 V pulse amplitude). The error bars show standard experimental deviations based on ten independent devices.

microseconds. The TS behavior originates from the field-induced Ag diffusion and redox reaction. When the power is turned off, Ag CFs break and diffuse to their minimum energy positions, inducing the relaxation to HRS. Repeatable relaxation features of 50 cycles indicate its good reproducibility. Spontaneous decay processes of the device after the set process are depicted in Fig. 2e. The experimental data were fitted with a stretched-exponential function, indicating a uniform relaxation property with decay time $\tau$ of 35 ± 3 ms.

To systematically study how the relaxation property is influenced by the internal dynamics, the electrical pulse response of FLBP–CsPbBr₃ TSM was detected. As shown in Fig. 2f, under

applied electrical pulses with different amplitudes from 0.6, 0.8, 1.0, 1.2 to 1.4 V and fixed duration of 20 ms, the TSM showed a sudden increase of current to Icc level of $10^{-5}$ A and the current was relaxed to the HRS within the limited decay time when the applied voltage was removed, suggesting a typical short-term memory feature. It is worth pointing out that a long interval time of 0.15 s was employed in between these single pulse measurements to ensure that the TSM has enough time to decay back to its original HRS[44,45]. Here, the switching voltage of 0.6 V was lower than the $V_{th}$ in Fig. 2a, because the programming operation with a relatively long time can induce resistance transition with a lower voltage (Supplementary Fig. 6). Statistics analysis of

relaxation time with respect to the voltage amplitude from 0.6 to 1.4 V and pulse width from 10 to 50 ms is shown in Fig. 2g and Supplementary Figs. 7 and 8. The voltage with higher amplitude and longer duration leads to a longer relaxation time with a narrow distribution. The stochasticity of relaxation behavior is correlated to the random growth and rupture of Ag CFs.

**TSM-based biomimetic compound eye under the fixed stimuli.** The artificial vision system of industrial and military robot applications is expected to show a wide FoV-detection capability for notifying an impending collision[43]. In contrast to the human eye, the compound eye contains thousands of integrated optical units named ommatidia which are spherically assembled along a curvilinear surface with each ommatidium point pointing toward a different direction. The omnidirectionally arranged ommatidium collects incident photons independently with a narrow angular acceptance which contributes to the wide FoV-detection capability (Fig. 3a). Here, the biomimetic compound eye with hemispherical retina made of $20 \times 20$ FLBP–CsPbBr$_3$ TSM flexible arrays on the polydimethylsiloxane (PDMS) substrate is demonstrated (Fig. 3e). To investigate the production yield of the FLBP–CsPbBr$_3$ TSM device-based crossbar array, we measured all the 400 devices in a $20 \times 20$ crossbar array. The $I$–$V$ characteristics of the 400 memristor devices are provided in Supplementary Figs. 15–19, with 337 of them showing stable threshold switching characteristics and a production yield of 84.25%.

First, the optical uniformity and light-sensing capability of biomimetic compound eye to the light illumination with fixed distance were investigated. As shown in Fig. 3d, the systematic distribution with sharp power of focusing spots indicates that the artificial compound eyes have high optical uniformity along $x$ direction and $y$ direction. It is worth noting that the light beam is transmitted both through the ommatidium and between over all the dome regions. The angle-resolved photocurrent of a single FLBP–CsPbBr$_3$ TSM device is shown in Fig. 3f, h. During the measurements, 365-nm UV-light pulses with laser power of 0.72 mW, the pulse width of 10 ms, and reading voltage of 0.1 V were employed. The azimuthal angles of the light source are varied by rotating the focusing optics. The device conductance sharply increases under the light pulse stimulus and the photocurrent of FLBP–CsPbBr$_3$ TSM exhibits a highly directional response. The current values are $2.765 \times 10^{-5}$, $1.965 \times 10^{-4}$, $2.614 \times 10^{-4}$, $3.628 \times 10^{-4}$, $3.360 \times 10^{-4}$, $2.221 \times 10^{-4}$, and $1.166 \times 10^{-4}$ A for the UV-light source with the incident angle of 0, 40, 60, 90, 110, 130, and 150°, respectively. The strongest photocurrent signal is obtained when the incident light is positioned at 90°. With the hemispherical shaped biomimetic compound eye demonstrated in Fig. 3e, the identical incident angle of 180° along both the $x$ and $y$ direction is regarded as the FoV for our artificial apposition compound eye. Apart from the light-power sensing behavior demonstrated in Fig. 3b, the wavelength-dependent effect of threshold switching characteristics improves the essential light sensitivity and color-selectivity for the artificial vision system, which would improve the perception accuracy of the system. The $I$–$V$ characteristics of FLBP–CsPbBr$_3$ TSM under light irradiance with different wavelengths are shown in Fig. 3c. As wavelengths decrease from 520 to 365 nm, the $V_{th}$ decreases from 1.081, 0.997, 0.890 to 0.296 V while the off-current of devices varies from $5.984 \times 10^{-11}$, $1.823 \times 10^{-10}$, $6.207 \times 10^{-10}$ to $8.882 \times 10^{-10}$ A. The interactive contribution of wavelength and incident angle to photocurrent signal is demonstrated in Fig. 3g, i, which were further mapped into Supplementary Fig. 20, suggesting successful wavelength discrimination of FLBP–CsPbBr$_3$ LGMD neuron-based artificial compound eye in the UV–visible region.

The underlying mechanism of the light-sensing capability of FLBP–CsPbBr$_3$ TSM was investigated by the Kelvin probe force microscopy (KPFM) measurements. The surface potential characteristics were evaluated under a UV illumination. For comparison, the light irradiance with different intensities was applied to the FLBP–CsPbBr$_3$ layer. As displayed in Supplementary Fig. 12, the surface potential steadily increases from 69.2, 134.0, 211.7 to 273.2 mV when the light power increases from 0.00, 0.24, 0.72 to 2.50 mW. Significant surface potential enhancement was observed in the FLBP–CsPbBr$_3$ layer since the light illumination on the CsPbBr$_3$ QDs film can trigger the generation of photocarriers. The photogenerated electrons can be easily transferred from CsPbBr$_3$ to FLBP through the internal electric field and thus the photo-induced holes are left in the valence band of the CsPbBr$_3$ which induces the increase of surface potential.

**TSM-based biomimetic compound eye under the looming stimuli.** Next, the response of FLBP–CsPbBr$_3$ TSM-based biomimetic compound eye to approaching stimuli was further investigated. Figure 4a shows the scenario in which a bird predator is approaching the locust. The looming object perceived by the locust at different distances from position A1 to position A4 can be mimicked by the ramping of the power of the static light source. Therefore, collision avoidance response of FLBP–CsPbBr$_3$ TSM-based biomimetic compound eyes are characterized with the experimental setup as shown in Fig. 4c. During the measurements, a voltage stimuli train consisting of +0.2 V pulses, the interval of 50 μs, and duration of 50 μs was employed to activate the single TSM device. It is worth pointing that the applied electrical pulses are programmed with a relatively low amplitude and long interval time to avoid the sudden formation of Ag CFs. The light power of the applied laser was ramped from 0.00 to 2.50 mW with time intervals of 0.32 s to determine the increased power rate (Supplementary Fig. 21). Note that the collision takes place when the applied laser reaches its peak power at 2.50 mW in Fig. 4b. As shown in Fig. 4d, the conductance of FLBP–CsPbBr$_3$ TSM increases, peaks, and decreases as the laser power increases. The increasing tendency and subsequent decreasing trend are analog to the excitatory and inhibitory response of LGMD neuron, respectively (Fig. 4e, g). The combination of excitatory and inhibitory response emulates the escape response of LGMD neuron as collision becomes imminent. Interestingly, the current peak occurs before laser power reaches its peak value so that the FLBP–CsPbBr$_3$ TSM is promising to be employed as an impending collision indicator to detect the collision before impact.

The optical modulation of the TSM conductance can be clarified by the temporal heat summation effect. Our FLBP–CsPbBr$_3$ TSM exhibits a pronounced second-order memristor effect in which the size of Ag CFs is the first-order state variable and the local temperature of TSM, temperature ($T$) is the second-order state variable. For electrical pulses operation of TSM, when the interval of electrical pulses is long or the amplitude of electrical pulses is low, Joule heating created from the previous pulse could be partially dissipated[46–49]. Applying the extra number of pulses to increase the inside temperature of the device could accelerate the drift and diffusion of Ag$^+$. In our light-modulated FLBP–CsPbBr$_3$ TSM case, it should be noted that the value of second-order state variable $T$ is strongly influenced by the photon-induced internal electrical field. The results of optical spectroscopy (Supplementary Fig. 9) indicate that the light induces electron transport from the photo-excited CsPbBr$_3$ QDs to FLBP NSs[36,37]. Then the injected electrons will be trapped in FLBP owing to its large surface area and high

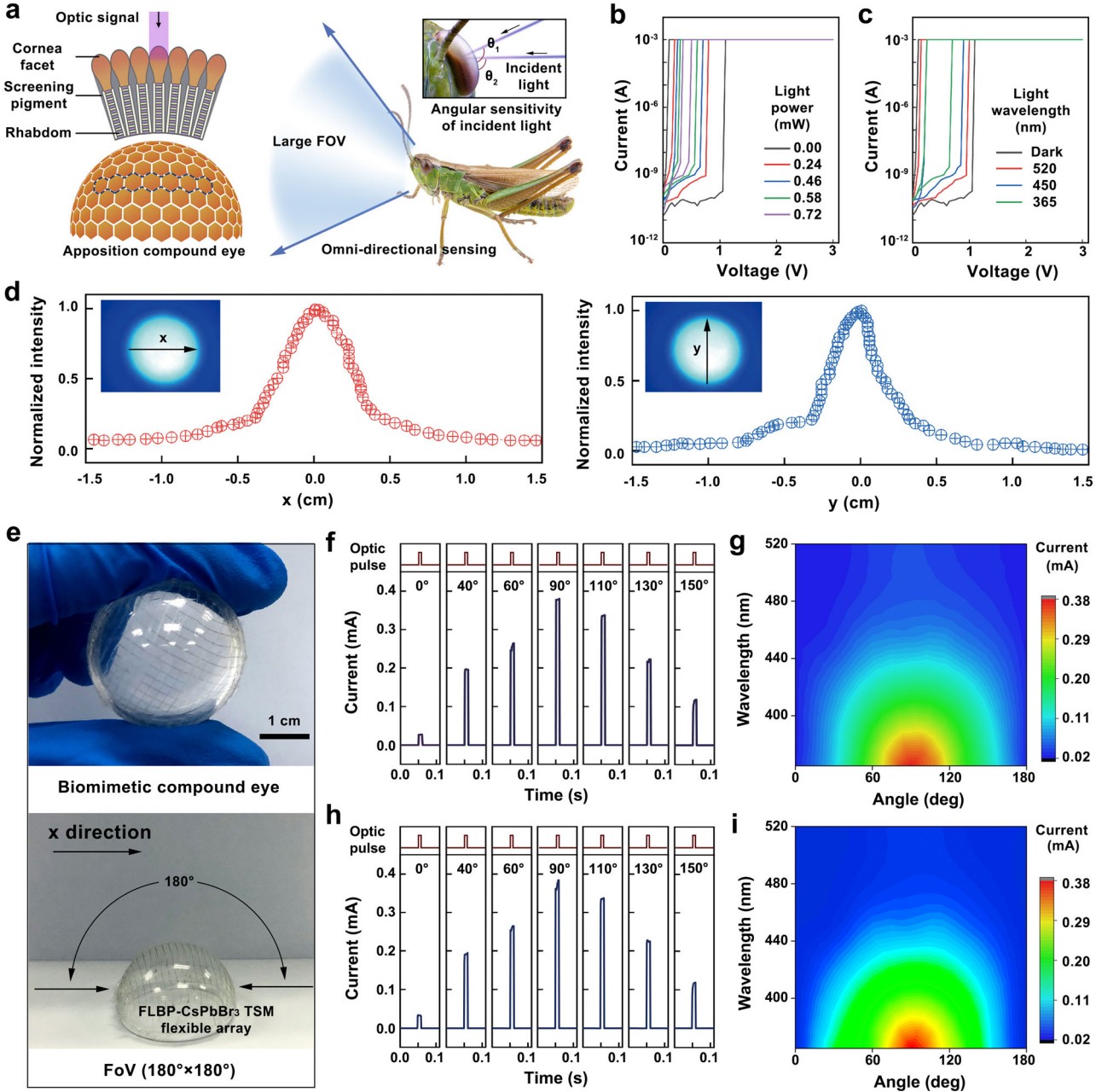

**Fig. 3 Biological and artificial apposition compound eye. a** Detailed structure of the apposition compound eye of the locust with omnidirectional optical sensing function. **b, c** I–V characteristics of FLBP–CsPbBr₃ TSM under positive sweep with different light power (**b**) and light wavelengths (**c**). **d** Measured angular dependence of the normalized power along the respective x and y direction. **e** Schematic of FoV of the device-level biomimetic compound eye. **f, h** Angle-sensitive responses in the TSM for optic pulsed (365 nm wavelength, 0.72 mW power, 10 ms pulse width) tests along x direction (**f**) and y direction (**h**). **g, i** Recorded respective device conductance with the different incident light along x direction (**g**) and y direction (**i**).

carrier mobility, leading to the accumulation of electrons in FLBP and the formation of the internal electrical field. The applied light irradiance could increase the device temperature for modulating the growth and rupture of the Ag CFs. When the looming stimulus is far from the TSM, the device receives limited amounts of photons with moderately increased temperature. Thus, the Ag drift process is accelerated to induce the formation of Ag CFs, leading to the increasing trend of conductance. When the approach object is almost on a collision course, considerable Joule heating would evoke the rupture of Ag CFs since the FLBP–CsPbBr₃ TSM absorbs more incident photons to build a high internal electrical field[50–53]. Therefore, the current of the device reaches its peak value and then exhibits a decreasing

tendency. The operation principle of our artificial LGMD neuron is depicted in Fig. 4f, h. Notably, an early switching event will be triggered at a higher temperature (Supplementary Fig. 22), which is also in consistent with the second-order memristor effect.

Concrete integration of a large FoV artificial vision system with the nonmonotonic response of TSM device was demonstrated by measuring the current values of FLBP–CsPbBr₃ TSM on the hemispherical visual system in a specific position from 0 to 90° under a series of electrical pulses in response to different laser ramp rates. Supplementary Fig. 24 illustrates that nine specific positioned TSM devices were selected with the light incident angle from 0 to 90°. During the measurements, a voltage stimuli train consisting of +0.2 V pulses, duration of 50 μs, and interval

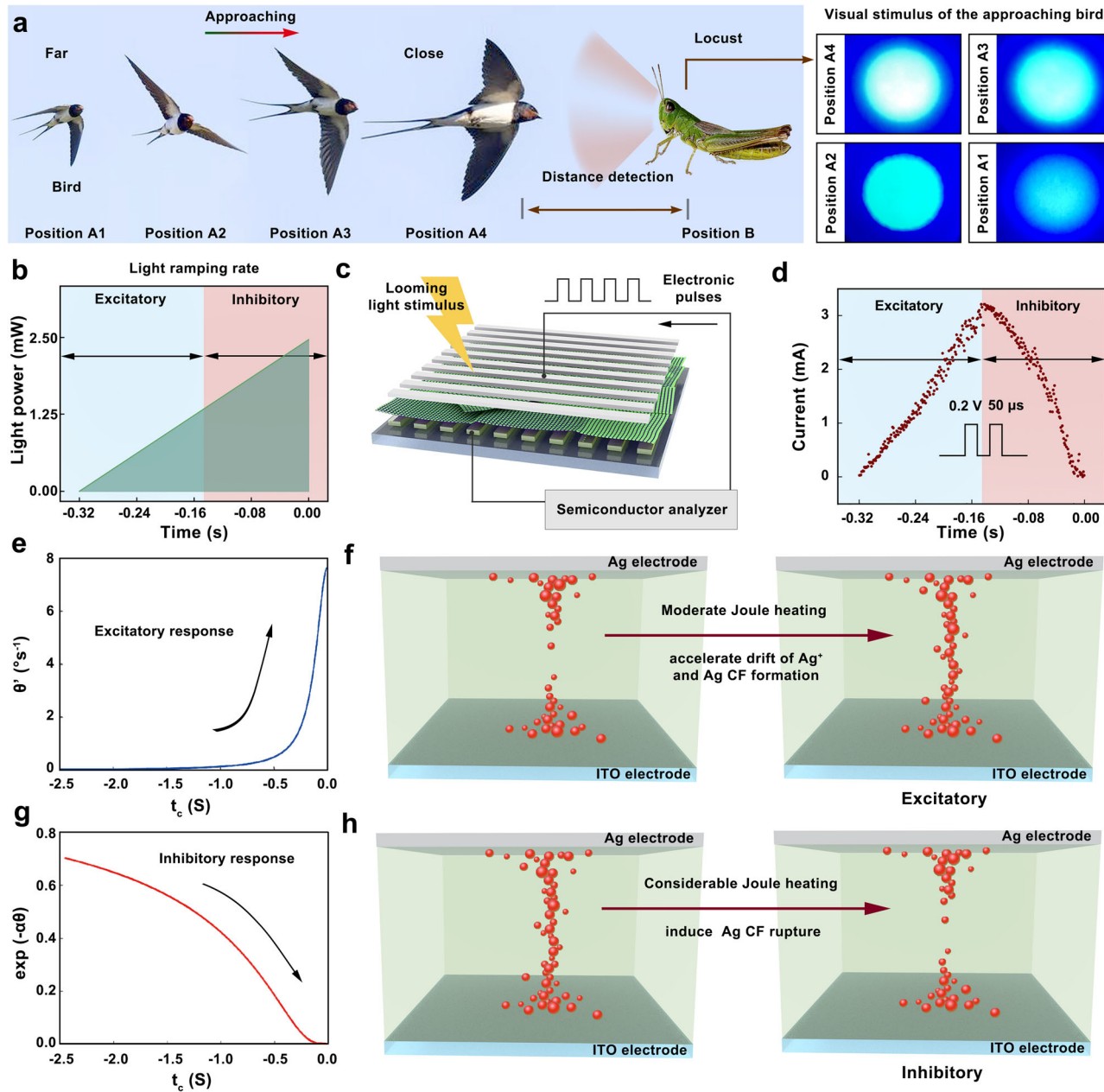

**Fig. 4 Artificial response of the FLBP–CsPbBr₃ TSM to looming stimuli. a** Schematic of approaching bird to the locust and snapshots of the stimulated visual information detected by the locust in the form of the monotonic increase in the optic power from position A1 to position A4. **b** Monotonic increase in the light power ramped from 0.00 to 2.50 mW determining the approaching speed of the object. **c** Schematics of the application of the programming electronic pulses to the top electrode of the FLBP–CsPbBr₃ TSM together with the looming light stimulus. The current response of the TSM is recorded by the semiconductor analyzer spontaneously. **d** Excitatory and the inhibitory response of the device to a looming light stimulus with simultaneously applied programming electronic pulses (0.2 V voltage pulse, 50 μs duration, 50 μs interval). **e** Excitatory response of the biological LGMD neuron in response to a direct approaching object before the impending collision. **f** Accelerated Ag drifts process to induce the formation of Ag-conductive filaments (CFs) in FLBP–CsPbBr₃ TSM with moderate Joule heating effect, leading to the excitatory response of the artificial LGMD neuron. **g** Inhibitory response of the biological LGMD neuron in response to a direct approaching object as collision becomes imminent. **h** Rupture of Ag CFs in FLBP–CsPbBr₃ TSM induced by considerable Joule heating effect, leading to the inhibitory response of the artificial LGMD neuron.

of 50 μs were employed. The ramp rate of laser light can be used to capture different looming speeds. The laser intensity was ramped from 0.00 to 2.50 mW with different time intervals to determine the increased intensity rate in each case. Obviously, the response of current is nonmonotonic for all the measurements (see the illustration of Fig. 5a and Supplementary Fig. 25). Figure 5b maps the inflection point values of TSM output current as per the data shown in Fig. 5a. As expected, the monotonic trend of the current value was obtained as a function of the object

looming speed for the given incident angles. The monotonical increment in the current value with the laser incident angles further demonstrates the most sensitive collision detection with the 90° incident angle. Therefore, the front position TSM on the flexible compound eye with the 90° incident angle offers optimal signal detection. In addition, the 3D spatial distribution of the time-to-collision detection ($\tau_D$) of our large FoV artificial vision system is demonstrated in Fig. 5c. The further optimizations of our flexible artificial vision system with incident light

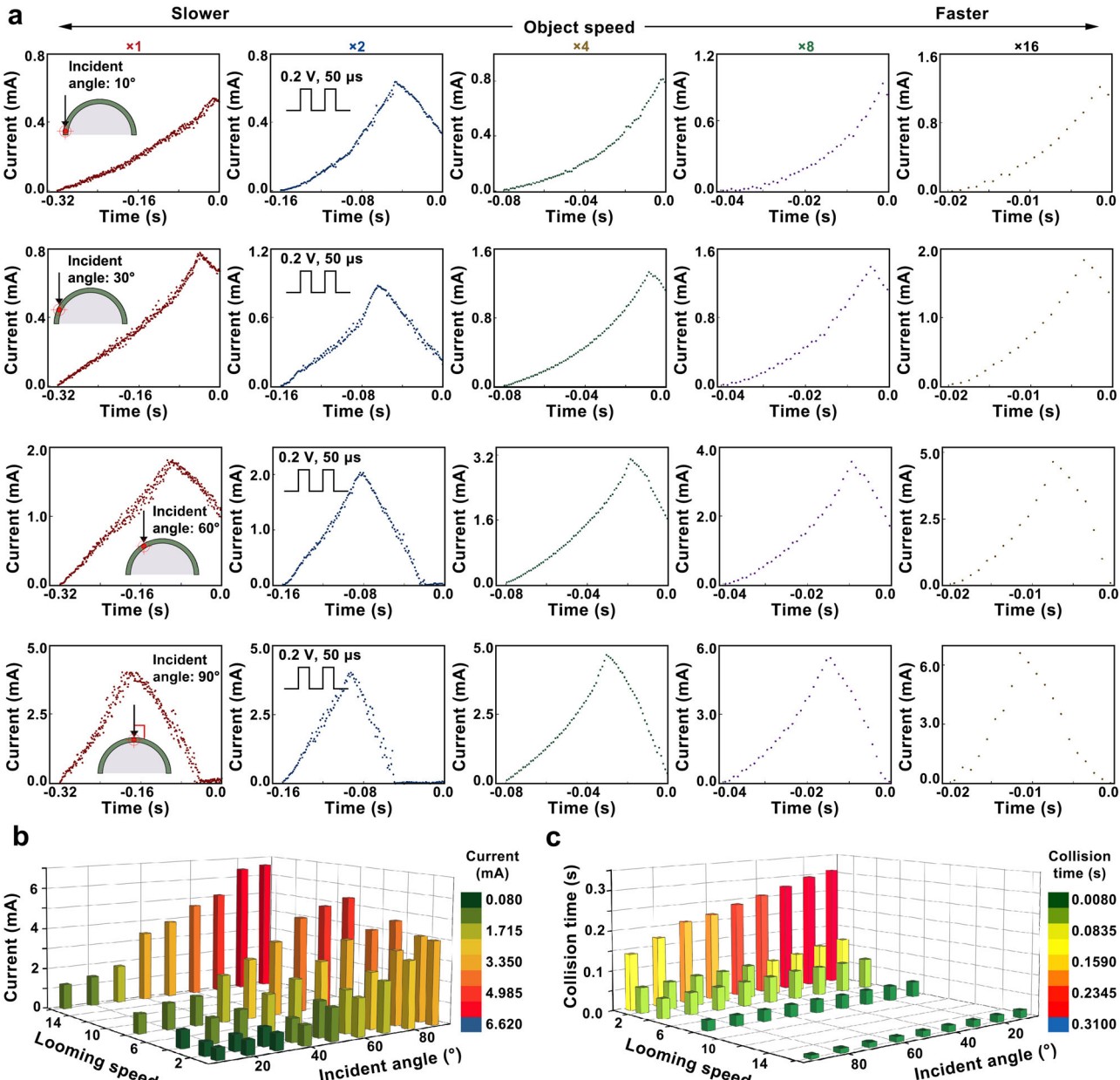

**Fig. 5 Escape response of FLBP–CsPbBr₃ TSM. a** Output current of the FLBP–CsPbBr₃ TSM in response to different looming object speed with different incident angles. The laser power ramp from 0.00 to 2.50 mW with different rate to represent the different looming object speed. **b** Inflection point values in the output current of the FLBP–CsPbBr₃ TSM for different looming object speeds or the different positioned device with specific incident angles. **c** Detected collision time before impact by the FLBP–CsPbBr₃ TSM for different looming object speeds or the different positioned devices with specific incident angles.

may potentially bring forth applications in the vision chip of autonomous robots for indicating collision.

**Artificial LGMD neurons for robot navigation.** Given the optical modulated TS behavior of FLBP–CsPbBr₃ TSM, the artificial LGMD neuron can be built by the simple circuit as shown in Fig. 6a and Supplementary Fig. 26, where the FLBP–CsPbBr₃ TSM is connected with a capacitor ($C_p$) in parallel and with a resistor in series ($R_1$)[54–56]. First, the spiking behavior of artificial LGMD neuron was investigated by applying a pulse train with an amplitude of 2 V, pulse width of 50 ms, and frequency of 10 Hz into the left node. The voltage on the capacitor was recorded as the output voltage. At the commencement of the input signal, the noticeable integration process of the capacitor

can be detected in Fig. 6b. Then the output voltage oscillates with fixed amplitude. This phenomenon can be explained as: the artificial neuron can be separated into the charging loop and the discharging loop. Owing to the voltage dividing effect, the $C_p$ can be charged initially via charging loop because the voltage mainly drops across the FLBP–CsPbBr₃ TSM device (we design the neuron with $R_{OFF} > R_1$, where $R_{OFF}$ is the high resistance of FLBP–CsPbBr₃ TSM at the initial state, ~500 TΩ). Once the voltage of $C_p$ reaches its threshold, the TSM will be switched from HRS to LRS ($R_{on}$~1 kΩ), and the capacitor discharges via discharging loop, inducing the firing of the artificial neuron[57–59]. The relationship of firing rate with respect to the value of $C_p$ and $R_1$ is shown in Fig. 6c, d as per the data shown in Supplementary Figs. 27 and 28. The bigger $C_p$ leads to the slower integration

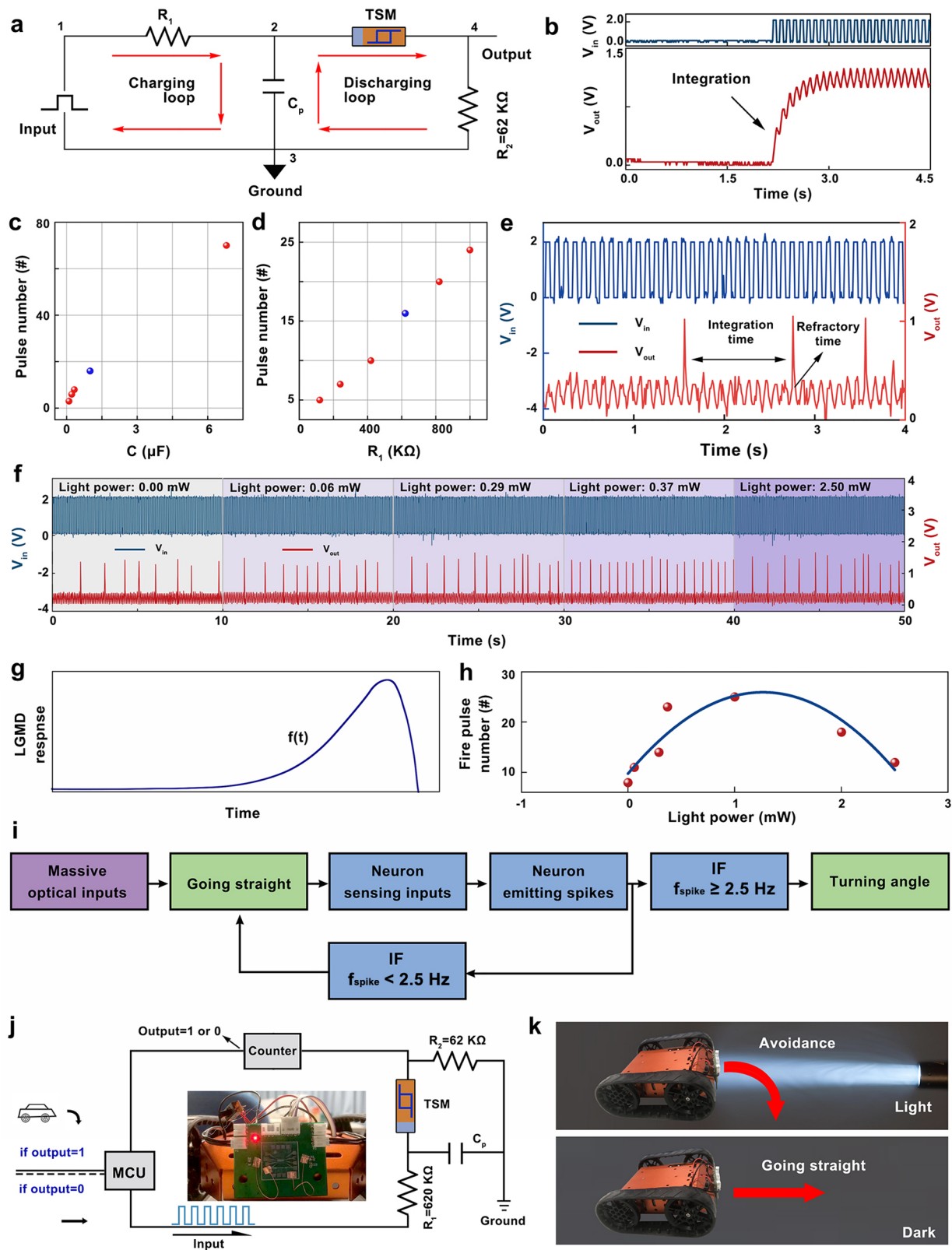

while a larger $R_1$ reduces the input current and induces the slow charging process, therefore delaying the firing time. According to the different $R_1$ implemented circuit conditions, we built the artificial LGMD neuron circuit with $R_1$ of 620 KΩ and the capacitor of 1 μF for the following neuron behavior analysis. As shown in Fig. 6e, the neural refractory period is emulated with input voltage pulse through the LRS TSM device, which virtually zero the net charge integration in the capacitor. After the complete refractory period, TSM device switched from LRS back to HRS with the reduced output voltage from the firing $V_{th}$ to the resting voltage. The firing frequency is highly dependent on the voltage stimulus magnitude and frequency (Supplementary Fig. 29). In addition, we investigated the writing energy of the device for a single-voltage pulse by measuring the current

**Fig. 6 Optic signal processing. a** Circuit of an artificial neuron constructed with the FLBP–CsPbBr$_3$ TSM. **b** Voltage integration behavior of the capacitor in the neuron circuit in (**a**). **c, d** Corresponding pulse number (2 V pulse magnitude, 50 ms pulse width) for the different capacitors to be 100% charged (**c**) under different R$_1$ implemented circuit conditions (**d**). The blue spot represents the selected capacitor and R$_1$ for the constructed circuit of the artificial neuron. **e** Corresponding refractory period and integration moment to trigger neuronal firing. **f** Light-shifted neuron firing probabilities with laser power varies from 0.00 to 2.5 mW. **g** Schematic diagram illustrating the time evolution of $f(t)$ functioned LGMD response. **h** Historical statistics of the fire pulse number with different light power varies from 0.00 to 2.5 mW. **i** Decision flow diagram of the motion trajectory from the robot car with the artificial neuron circuit. **j** Schematic illustration of the model car test setup. The inset shows the back-view photo of a robot car with mounted FLBP–CsPbBr$_3$ TSM on a printed circuit board. **k** Schematic decision-making for the robot car with optic signal processing ability.

between two terminals (Supplementary Fig. 30). The device exhibited an energy consumption of 20.3 nJ for the 50 ms width pulse, which was further reduced to 2.1 nJ under a pulse width of 50 μs.

The LGMD neuron can respond to optical input and encode them into spikes. The light-mediated spike behaviors were measured by applying the fixed pulse train with a single pulse amplitude of 2 V, the interval of 50 ms, and pulse width of 50 ms under UV-light irradiance of different power (0.00–2.50 mW). As shown in Fig. 6f, the spiking frequency ($f_{spike}$) strongly depends on the power of light. According to the previous results, the conductance of FLBP–CsPbBr$_3$ increases, peaks, and decreases as the power of UV illumination increases (Fig. 6h). The $f_{spike}$ follows a similar trend as the increase of UV irradiance which is analog to the response of LGMD neuron to a looming object (Fig. 6g), and reaches a 2.5 Hz (25 fired pulse within 10 s) to detect the subsequent collision. Whereas the influence of light wavelength on the $f_{spike}$ is demonstrated in Supplementary Fig. 31. The $f_{spike}$ decreases as the light wavelength shift from 365 to 520 nm.

To demonstrate the potential integration of artificial LGMD neurons with circuits, the $f_{spike}$ controlled systems is proposed to accomplish the collision avoidance function of the robot car. The decision-making process for the robot car in the motion trajectory is explained in Fig. 6i, j and Supplementary Figs. 32 and 33. In addition to Fig. 6k, Supplementary Movies 1–3 also demonstrate the process of the robot car approaching the imminent stimulus. According to the characteristics of the LGMD neuron model in Fig. 6h, its avoidance behavior should appear at the highest point, which is defined as 2.5 Hz. By using a counter to count the fire pulse number within 10 s to obtain $f_{spike}$ compared with 2.5 Hz, the microcontroller can be driven to achieve the output of 0 or 1. As the robot car approaches the light source, the $f_{spike}$ of the LGMD neuron increases. When $f_{spike}$ reaches 2.5 Hz, the robot car will turn an angle to avoid imminent collision. Conversely, the robot car will move linearly with a $f_{spike}$ less than 2.5 Hz. Thus, the output $f_{spike}$ controls the turning time of the robot car, which derives a directional decision for the LGMD neuron under different situations. This proof-of-principle demonstration of the fully integrated TSM neuron can be expanded to implement learning systems of higher complexity in an energy-efficient manner.

## Discussion

We have reported a biomimetic compound eye based on artificial LGMD neuron with wide FoV-detection capability (180° × 180°) and nonmonotonic collision avoidance response by employing 20 × 20 two-terminal FLBP–CsPbBr$_3$ TSM crossbar. The conductance of FLBP–CsPbBr$_3$ TSM increases, peaks, and decreases as the light power increases which is analog to the excitatory and inhibitory response of LGMD neuron toward the looming object. The optical modulation of the TSM conductance can be clarified via the temporal heat summation effect. When the looming stimulus is far from the TSM (low irradiance), the Ag drift process is accelerated to induce the formation of Ag CFs. While when the

approach object is almost on a collision course (high irradiance), considerable Joule heating would evoke the rupture of Ag CFs. Given the optical modulated TS behavior of FLBP–CsPbBr$_3$ TSM, the artificial LGMD neuron can be built by the simple RC circuit. The LGMD neuron can respond to optical input and encode them into spikes with the nonmonotonic response. Furthermore, LGMD neuron was also integrated with circuits to accomplish the collision avoidance function of robots. We believe that the LGMD neuron based on a compact TS memristor can potentially lead to the advance of area-efficient, low-energy, and low-cost vision chips for applications in autonomous robotics designs.

## Methods

**Materials preparation.** Bulk BP crystal as raw material, which was purchased from Nanjing MKNANO Tech. Co. Ltd (http://www.mukenano.com). PDMS precursor was purchased from Dow Corning. Lead bromide (PbBr$_2$, 99.999%), barium carbonate (Cs$_2$CO$_3$, 99.999%), hexane (anhydrous, 98%), 1-octadecene (ODE, technical grade, 90%), toluene (anhydrous, 98%), oleic acid (OA, technical grade, 90%), oleylamine (OLA, technical grade, 70%), methyl acetate (MeOAc, anhydrous, 99.5%), and N-methyl pyrrolidone (NMP, AR) were purchased from Sigma-Aldrich.

**Synthesis of CsPbBr$_3$ QDs.** The Cs-oleate precursor was synthesized by loading 0.81 g of Cs$_2$CO$_3$, 40 mL of ODE, and 2.5 mL of OA to a 3-neck flask (100 mL), and heating to 120 °C for 1 h under nitrogen atmosphere to ensure complete reaction. Then the Cs-oleate precursor was maintained at 160 °C to avoid further precipitation. 0.188 mmol of PbBr$_2$ mixed with 5 mL of ODE was placed in another 3-neck flask under vacuum, and the temperature was maintained at 120 °C for 60 min. Then 0.5 ml of OA and 0.5 ml of OLA were injected into the 3-neck flask with raised temperature to 140 °C under nitrogen flow. 0.4 mL of the previous Cs-oleate precursor was Immediately injected into the flask. The flask with the fluorescent green-colored solution was immersed in an ice/water bath for quenching (5 s). Subsequently, the purification of CsPbBr$_3$ QDs was performed by centrifuging the as-synthesized CsPbBr$_3$ QDs with MeOAc solution at 9063.9 × g for 5 min. Then the redispersed CsPbBr$_3$ QDs in 1 mL MeOAc and 1 mL hexane were reprecipitated at 9063.9 × g for 5 min. The resulting CsPbBr$_3$ QDs solution was prepared by dispersing the QDs in 2 mL of anhydrous toluene.

**Synthesis of FLBP NSs.** In total, 100 mg of bulk BP crystal was immersed in NMP to obtain BP solution with a concentration of 1 mg/mL. With an ice bath inside an ultrasonic disruptor, the BP solution was sonicated for 12 h, the resulting FLBP NSs were obtained via the "liquid cascade centrifugation" process (4028.4–9063.9 × g).

**Preparation of self-assembled CsPbBr$_3$ QDs on FLBP NSs.** The facile self-assembly method was adopted to synthesized the FLBP–CsPbBr$_3$ nanocomposites in toluene. CsPbBr$_3$ QDs solution with different concentration (0.5, 1.0, 1.5, and 3.0 mg/mL) was added into the FLBP NSs solution with 0.5 mg/mL concentration. The FLBP–CsPbBr$_3$ nanocomposites were obtained after the following bath sonication at room temperature for 10 min.

**Fabrication of FLBP–CsPbBr$_3$ TSM.** PDMS precursor was first mixed with the curing agent (10:1 of the weight ratio) and coated onto a clean glass slide (4 × 4 cm size). The curing temperature was controlled at 70 °C in a vacuum drying oven. After 2 h curing, the peeled PDMS film was resized into 2 × 2 cm by cutting. For the preparation of the bottom electrode, 100 nm of the ITO layer was sputtered onto the PDMS substrate via magnetron sputtering technique. The FLBP–CsPbBr$_3$ film was prepared by spin-coating the FLBP–CsPbBr$_3$ solution onto the ITO-PDMS substrate with a speed of 3000 rpm. The subsequent annealing process was carried out at 100 °C in a high vacuum furnace for 30 min. The top Ag electrodes were prepared via the shadow mask-assisted thermal evaporation method.

**Electrical characterizations**. The existence of the FLBP–CsPbBr$_3$ was verified with the Powder X-ray diffraction (XRD) results using a Philips X'pert X-ray diffractometer (Cu Kα radiation). SEM images were obtained on a field emission scanning microscope (Bruker XFlash 6|10). The distributions of the CsPbBr$_3$ QDs on the FLBP NSs were analyzed using the high-resolution TEM (Tecnai F30). Optical characterizations of the FLBP film and CsPbBr$_3$ film were conducted on a UV-vis spectrophotometer (Agilent Cary 60) and Edinburgh Instruments (FLS 920). The electronic performance test bench in this manuscript includes a multi probes station and a Keithley 4200A-SCS parameter analyzer. The electric performance of the TSM device is characterized by an Agilent B1500A semiconductor parameter analyzer. In the TSM neuron circuit experiment, the input voltage column is generated by KEYSIGHT B2902A parameter analyzer, and the output resistance and the voltage potential on the capacitance are monitored at room temperature by the Tektronix TDX 2022C oscilloscope.

## Data availability

The data that support the plots within this paper and are available from the corresponding author upon reasonable request.

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

## Acknowledgements

This research was supported by the NSFC Program (grant nos. 62074104, 61974093, 52103297, and 62104154), Guangdong Province Special Support Plan for High-Level Talents (grant no. 2017TQ04X082), Guangdong Provincial Department of Science and Technology (Grant nos. 2018B030306028, 2019A1515110702, and 2020A1515011425), the Science and Technology Innovation Commission of Shenzhen (Grant nos. JCYJ20180507182042530, JCYJ20180507182000722, 20200804172625001, RCYX20200714114524157 and JCYJ20180305124214580), and the Beihang Hefei Innovation Research Institute (Project No. BHKX-19-02).

## Author contributions

Y.W. and Y.G. performed the experiments on TSM, fabricated the devices, analyzed the data, and wrote the paper. S.M.H. and X.C.X. designed and made the car to avoid obstacles model. J.J.W. and J.Q.Y. performed the IF model measurements. Z.Y.L., G.H.Z., and Y.Z. performed the PL mapping measurements. S.T.H. conceived and supervised the project and finalized the paper. All authors discussed the results and revised the manuscript.

## Competing interests

The authors declare no competing interests.
