## [Peer Review File · Nature Communications]

Memristor-based biomimetic compound eye for real-time collision detectionREVIEWER COMMENTS

Reviewer #1 (Remarks to the Author):

In their manuscript titled 'Light-mediated threshold switching memristor for biomimetic Lobula giant movement detector neuron', Y. Wang et. al. have mimicked the functionality of a collision detector neuron found in locusts called LGMD using a two-terminal threshold switching memristor (TSM) with structure of Ag/few layer black phosphorous nanosheets (NSs)-CsPbBr₃ perovskite quantum dots (QDs) heterostructure (FLBP-CsPbBr₃)/indium tin oxide (ITO). The light responsive TSM shows a non-monotonic response to increasing light intensity, similar to the functioning of LGMD neuron. This is achieved by the Ag conductive channel formation with increasing number of photons getting absorbed, which will also cause Joule heating. When the internal temperature crosses a certain value, this channel ruptures leading to decrease in conductance. This TSM is then integrated with an RC circuit to create spikes in which light induced conduction change will result in the modulation of spiking rate, thereby encoding light information to spikes.

While interesting, the reviewer believes that the manuscript is very similar to the work by Jayachandran et al. Nature Electronics 3 (10), 646-655, 2020. Even some of the figures in the manuscript have close resemblance with the above referenced article. The only difference is the demonstration of wide field of view (FoV) artificial vision system. Therefore, it remains unclear at this point whether this manuscript can meet the novelty requirement for Nature Communications. A concrete integration of a large FoV artificial vision system with its previous section consisting of collision avoidance is missing. The reviewer believes that putting more emphasis on this aspect and associated demonstrations can significantly improve the quality of the manuscript and might mitigate the novelty concern depending upon the rigorousness of the demonstration. Please see below for other suggestions/comments:

- 1) The key component of the manuscript is the light sensing capability of TSM. But after explaining the electrical performance, the focus goes directly to the response of FLBP-CsPbBr₃ TSM to looming stimuli. Some more characterization of the photoresponse behavior under a constant light source and explanation of the underlying mechanism may help in enhancing the quality of the manuscript. The results of optical spectroscopy indicating electron transport from photo-excited CsPbBr₃ QDs to FLBP NSs and trapping mechanism in FLBP is not clear, necessitating a detailed explanation of the same.
- 2) TSM rely on Joule heating for channel formation, irrespective of whether it is from electrical or optical pulses. In a practical operation scenario, how will the external temperature affect the performance of collision avoidance?
- 3) The demonstration of collision avoidance is interesting and shows the capability of the mechanism. Authors term the collision avoidance demonstration as a consequence of the 'unsupervised spike neural network (SNN)'. How is the system unsupervised? What are the underlying algorithms? How does the underlying neural network learn from the non-monotonic spiking output of TSM? In second paragraph, page 9, it is written that, "In case of straight moving, as the robot car becomes close to the light source, the spiking frequency of LGMD neuron increases and robot car at a constant speed will turn an angle to avoid the looming collision". How is the non-monotonicity of spiking output from TSM used for collision avoidance if the car turns just with increasing spiking frequency? It is strongly encouraged to provide a detailed explanation of Fig. i, j and supplementary figures 13, 14 and the underlying algorithms of collision detection.
- 4) Having a flexible photosensitive array will certainly help in creating hemispherical vision system, which allows achieving large FoV artificial vision systems. But a concrete integration of a large FoV artificial vision system with its previous section consisting of collision avoidance is missing. The reviewer believes that putting more emphasis on this aspect and associated demonstrations can significantly improve the quality of the manuscript.
- 5) As mentioned in the previous comment, the hemispherical artificial vision system is the most interesting aspect of this manuscript. In the section titled "Biomimetic compound eye based on artificial LGMD neuron", after explaining benefits of having a hemispherical artificial vision system, the wavelength-dependency is discussed. The reviewer struggles to understand how this paragraph can be part of this section.
- 6) A through explanation of Figure g and i will be helpful. Different wavelength radiation will invoke different threshold shifts. Also, for a given wavelength radiation, different devices at

different angles in the hemispherical array will respond differently. How can the system differentiate between wavelength dependent response and angle-dependent response?

7) "Apart from the light intensity sensing behavior demonstrated in Fig. 6b, the wavelength-dependent recognition improves the classification accuracy of various objects" – What kind of classification accuracy are authors referring to?

8) What is VF and how is it determined?

9) Natural compound eyes like apposition and refracting/reflecting superposition compound eyes comes with corneal lens to focus light onto the rhabdom. Is it possible to integrate optical lenses with the FLBP-CsPbBr₃ TSM flexible array?

10) Authors are encouraged to reduce spelling mistakes in the manuscript. For example 'TMS' in Figure 6e, and 'LGMD respnse' in Figure 5g. These are not critical and not deciding factors for considering the manuscript for publication, but still can provide a better impression for the reader.

Reviewer #2 (Remarks to the Author):

In this paper, the authors reported an lobula giant movement detector (LGMD) neuron based on a light-mediated threshold switching memristor. The light-dependent escape response of the LGMD neuron and its mechanisms were systematically demonstrated. Furthermore, the light-dependent properties of the LGMD neurons were implemented in robot navigation with obstacle avoidance and artificial eyes with wide detection functions. In my view, this work is interesting but some issues should be solved for further improvement.

1. The authors used BP and halide perovskite of CsPbBr₃ quantum dots to fabricate the two-terminal memristors. Why did the authors specifically used BP and CsPbBr₃? Did the authors consider the band alignment of the BP and CsPbBr₃ in material selection? Also, the authors should explain the electrons and holes transport under the light illumination and the effect of generated electrons and holes in terms of device operation and performance.

2. For automotive applications, the device should show stable and consistent characteristics under high temperatures. However, as the operation mechanism of the suggested FLBP-CsPbBr₃ TSM is based on Joule heating, the temperature of the device might highly affect the switching characteristics of the device. The authors should provide the electrical characteristics under diverse temperatures to ensure their applicability.

3. Another concern is the power consumption of the suggested device. To achieve obstacle avoidance, voltage pulses with long duration (50 ms) and interval (50 ms) should be consistently applied, which will lead to considerably high power consumption compared to previous reports on LGMD neurons (Nature Electronics 3, 646-655 (2020)).

4. In Figure 2b, the memristors show the threshold switching behavior with threshold voltage of about 1 V during DC voltage sweep. However, in Figure 2f, when a voltage pulse (0.6 V, 20 ms) is applied, the device showed threshold switching behavior. How is it possible to change the resistance of the device at a pulse amplitude smaller than the threshold voltage observed in DC voltage sweep?

5. The authors claimed that the voltage pulse with higher amplitude and longer duration leads to longer relaxation time with a narrower distribution (Figure 2g). Although the relaxation time increases with increasing amplitude and duration, the distribution seems to be irrelevant with the condition of voltage pulse.

6. In Figure S7, the authors claimed that the electrical properties of the device were changed depending on the concentration of CsPbBr₃. However, the explanation of the characteristics is insufficient. Also, in inset images, the thin film characteristics such as grain size, film quality, and roughness seem to be different. Did these film characteristics affect the electrical properties? In addition, information on the film thicknesses of each concentration should be presented for accurate comparison.

7. The statement "The FLBP-CsPbBr₃ TSM initially exhibited a resistance of $6.25 \times 10^{-11} \Omega$ at HRS" on page 2 seems to be wrong. The current of TSM at HRS is about 10⁻¹¹ A under 0.1 V, which leads to resistance over 10¹⁰.

8. The authors demonstrate the 20 × 20 TSM flexible array on the PDMS substrate for biomimetic compound eye. The authors should provide additional data to show the reliability of the array in a flexible environment.

Reviewer #3 (Remarks to the Author):

The authors present a novel study ranging from threshold switching memristor to artificial visual neuron. The proposed non-monotonic response of the device to light flow field could achieve bioinspired escape response with the implemented artificial neuron circuit, allowing an obstacle avoidance capability in robot navigation. The authors systematically investigated the influence of different looming object speed and voltage stimulus on output conductance so as to achieve the optimal collision detection. Additionally, biomimetic compound eye with hemispherical retina made of flexible device arrays showed the applicability of wide field-of-view detection. This original paper is clear in its theme, profound in its experimental and theoretical research. The topic will inspire further research in this field. Therefore, the reviewer recommends that the paper is accepted with major revisions on addressing points outlined below. These comments are not to criticize but to enhance the manuscript for publishing in high-impact Nature Communications.

1. Fixing device variability is a prerequisite for mass production. What about the device-to-device variability?
2. In Figure 2a, the authors demonstrated a symmetrical bidirectional threshold switching characteristic. Why are the negative polarity regions in Figure 2a nearly the same with the positive polarity regions? Please add discussion in the manuscript.
3. Given the voltage response of the TSM neuron evoked by repetitive pulses, firing probability in TSM neuron extracted from experiments under different input pulse magnitude and pulse frequency are suggested to be provided.
4. The devices were fabricated on a flexible PDMS substrate. What is the relationship between bending angle and device properties?
5. The authors claim that by comparing V_{spike} with V_F , the LGMD neuron can derive a directional decision from different situations. What is V_{spike} and V_F stand for? Please add discussion in the manuscript.
6. Given the large-scale deployment of neuromorphic computing in the future, it may be a good choice to take advantage of the mature silicon processing to be integrated with CMOS. The authors may state the implication of the current work for neuromorphic computing with considering CMOS compatibility.
7. The light intensity in Figure 6f,h is 0.72 mW, while Figure 6b use light intensity with the unit of mW/cm². Please unify the unit expression.

Point-by-Point Response to the Reviewers

Reviewer #1 (Remarks to the Author):

In their manuscript titled ‘Light-mediated threshold switching memristor for biomimetic Lobula giant movement detector neuron’, Y. Wang et. al. have mimicked the functionality of a collision detector neuron found in locusts called LGMD using a two-terminal threshold switching memristor (TSM) with structure of Ag/few layer black phosphorous nanosheets (NSs)-CsPbBr₃ perovskite quantum dots (QDs) heterostructure (FLBP-CsPbBr₃)/indium tin oxide (ITO). The light responsive TSM shows a non-monotonic response to increasing light intensity, similar to the functioning of LGMD neuron. This is achieved by the Ag conductive channel formation with increasing number of photons getting absorbed, which will also cause Joule heating. When the internal temperature crosses a certain value, this channel ruptures leading to decrease in conductance. This TSM is then integrated with an RC circuit to create spikes in which light induced conduction change will result in the modulation of spiking rate, thereby encoding light information to spikes.

While interesting, the reviewer believes that the manuscript is very similar to the work by Jayachandran et al. (Nature Electronics, 3, 646-655, 2020). Even some of the figures in the manuscript have close resemblance with the above referenced article. The only difference is the demonstration of wide field of view (FoV) artificial vision system. Therefore, it remains unclear at this point whether this manuscript can meet the novelty requirement for Nature Communications. A concrete integration of a large FoV artificial vision system with its previous section consisting of collision avoidance is missing. The reviewer believes that putting more emphasis on this aspect and associated demonstrations can significantly improve the quality of the manuscript and might mitigate the novelty concern depending upon the rigorousness of the demonstration. Please see below for other suggestions/comments:

ANS: Thank you for reviewing our paper. We appreciate your insightful comments on our research. We have revised the manuscript according to your suggestions and believe that these revisions have improved the paper.

First, the differences and novelty of our work are highlighted as below:

- (1) Device structure: In this nature electronics paper (ref 29), the device is based on the three-terminal structure with multiple layers including floating gate, dielectric layer and photoresponse layer, back gate and source-drain electrodes. While in our work, we adopted two-terminal threshold switching memristor with simple and compact structure which can be easily integrated with other chips.
- (2) Operation mode: In nature electronics paper (ref 29), the MoS₂ photodetector responding to the looming stimuli exhibits excitatory nature while the non-volatile charge trapping capability of the floating-gate memory induces an inhibitive response. Integrating both light stimuli and back gate pulse programming ensures the successful emulation of escape behavior of a LGMD neuron. The operation mode of this device is relatively complicated in which the additional electrical pulses should be applied on the back gate, while in our

work, the non-monotonic current response can be triggered only by the visual stimuli which is more analogue to the biological LGMD neuron.

- (3) Physical principle: In nature electronic paper (ref 29), the charge trapping in the floating gate (gate-screening effect) induces the shift of threshold voltage and inhibitory of current while the photovoltaic effect of MoS₂ induces the excitatory of current. While in our paper, the non-monotonic current response is originated from the photo-regulated Ag conductive filament formation/rupture in the memristor. The photo-induced Joule heating effect of second-order memristor is the main mechanism.
- (4) Prototype of artificial LGMD neuron and its application in robot navigation and artificial compound eyes: In nature electronics paper (ref 29), the author just employed the current response of their three terminal devices to emulate the escape response of LGMD neuron, the hardware implemented artificial LGMD neuron are missing. In our work, we built the artificial LGMD neuron by the simple RC circuit. In this hardware implemented LGMD neuron, the current response can be translated to firing rate which is more analogue to the biological LGMD algorithm. In addition, in our work, the robot navigation and artificial compound eyes with wide-field detection are demonstrated.

In summary, our work first reported an artificial LGMD neuron based on light-mediated threshold switching memristor with two-terminal structure and simple operation mode. This is the first time to adopt second order memristor effect to emulate the escape response of LGMD neuron. In addition, the robot navigation and artificial compound eyes with wide-field detection based on memristor-implemented LGMD neuron are first demonstrated.

Furthermore, concrete integration of a large FoV artificial vision system with the collision avoidance characterization was highlighted in the new Fig. 5 in the revised manuscript and Supplementary Fig. 24 in the revised Supporting Information. During the measurements, a voltage stimuli train consisting of +0.2 V pulses, duration of 50 μ s and interval of 50 μ s were employed. The artificial vision system can perceive varied light incident angle from 0° to 90° with different positioned TSM device while monitoring electrical signals simultaneously.

Please find below our responses (in blue) to each of your specific comments (in black). Revisions to the original article are indicated in red.

(1) The key component of the manuscript is the light sensing capability of TSM. But after explaining the electrical performance, the focus goes directly to the response of FLBP-CsPbBr₃ TSM to looming stimuli. Some more characterization of the photoresponse behavior under a constant light source and explanation of the underlying mechanism may help in enhancing the quality of the manuscript. The results of optical spectroscopy indicating electron transport from photo-excited CsPbBr₃ QDs to FLBP NSs and trapping mechanism in FLBP is not clear, necessitating a detailed explanation of the same.

ANS: In order to characterize the photoresponse behavior under a constant light source and explain the underlying mechanism, we evaluated the surface potential via the KPFM measurements. Light irradiance with different intensities (0.00 mW, 0.24 mW, 0.72 mW and

2.50 mW) was applied to the FLBP-CsPbBr₃ layer. The corresponding *I-V* characterization of the light responsive TSM with varied light powers (0.00, 0.24, 0.46, 0.58 and 0.72 mW) are provided in Fig. 3b with the same voltage sweeping of 0-3 V. The ramping power of the UV light source can effectively reduce the threshold voltage from 1.099, 0.804, 0.697, 0.599 to 0.497 V, and gradually increase the current density of the device in HRS state from 7.229×10^{-11} to 9.176×10^{-10} at 0.3 V. We added the relevant information to the manuscript and the Supporting Information as below.

In the manuscript:

cc

Figure 3b. *I-V* characteristics of FLBP-CsPbBr₃ TSM under positive sweep with different light power.

The underlying mechanism of light sensing capability of FLBP-CsPbBr₃ TSM were investigated by the Kelvin probe force microscopy (KPFM) measurements. The surface potential characteristics were evaluated under a UV illumination. For comparison, the light irradiance with different intensities was applied to the FLBP-CsPbBr₃ layer. As displayed in the Supplementary Figure 11, the surface potential steadily increases from 69.2 mV, 134.0 mV, 211.7 mV to 273.2 mV when the light power increases from 0.00 mW, 0.24 mW, 0.72 mW to 2.50 mW. Significant surface potential enhancement was observed in the FLBP-CsPbBr₃ layer since the light illumination on the CsPbBr₃ QDs film can trigger generation of photocarriers. The photogenerated electrons can be easily transferred from the CsPbBr₃ to FLBP through the internal electric field and thus photoinduced holes are left in the valence band of the CsPbBr₃ which induces the increase of surface potential.”

In Supporting Information:

“As shown in the Supplementary Fig. 11, the conduction band (valence band) of the few layered FLBP and CsPbBr₃ QDs is -4.58 eV (-4.67 eV) and -3.3 eV (-5.7 eV), respectively, hence the hybrid junction is expected to be type-I with a band offset of 1.28 eV (The electronic band structure of the FLBP varies with the layer numbers. Here we take 5 layered FLBP as an example). The large band offset induces the separation of the photogenerated electron-hole pairs in CsPbBr₃ and transportation of the electrons to FLBP. Additionally, the photoinduced electron transfer from CsPbBr₃ to FLBP was further confirmed by the KPFM measurements in Supplementary Fig. 12.

Supplementary Figure 11. Sketch of the conduction and valence band profiles and the electron-hole dynamics under irradiation.

Supplementary Figure 12. Surface potential of the FLBP-CsPbBr₃ layer under different light illumination (fixed light wavelength: 365 nm) recorded by in situ AFM electrical nano-technology (scale bar, 100 nm). The lower panel shows the respective surface potential profile.”

2) TSM rely on Joule heating for channel formation, irrespective of whether it is from electrical or optical pulses. In a practical operation scenario, how will the external temperature affect the performance of collision avoidance?

ANS: To evaluate the temperature effect on the performance of the FLBP-CsPbBr₃ TSM, the threshold switching characteristics of device were measured at different temperatures. We provide the electrical characteristics of the device under diverse temperatures in the manuscript and Supporting Information as below.

In the manuscript:

“The operation principle of our artificial LGMD neuron is depicted in the Fig. 4f, h. Notably, an early switching event will be triggered at a higher temperature (Supplementary Fig. 21), which is also in consistent with the second-order memristor effect.”

In Supporting Information:

“The substantial output signals shown in Supplementary Fig. 21 suggests the onset of a switching event. It was discovered that at a higher temperature, the early switching event will be triggered. When the temperature was increased from 298 K to 373 K, the amplitude of the current signal increased while the switching time of the signal decreased.

Supplementary Fig. 21 plot temperature dependent characteristics of devices by applying fixed 0.2 V voltage pulses (50 μs in width, 50 μs in interval). The current increases with temperature in the pre-switching state, suggesting a thermally activated electron hopping mechanism (*Nat. Commun.*, **2018**, 9, 417). After occurrence of switching event, the current value decreases as the temperature increases, suggesting that the conduction mechanism follows a metallic behavior. This result implies that there would be residual Ag clusters existed between Ag electrode and ITO electrode to reduce the effective gap distance, which can effectively engineer the conduction following a thermal-assistant hopping mechanism.

Supplementary Figure 21. Excitatory and inhibitory response of the device to a looming light stimulus with simultaneously applied programming electronic pulses under diverse temperatures ranging from 298 K to 372 K (0.2 V voltage pulse, 50 μ s duration, 50 μ s interval).”

3) The demonstration of collision avoidance is interesting and shows the capability of the mechanism. Authors term the collision avoidance demonstration as a consequence of the ‘unsupervised spike neural network (SNN)’. How is the system unsupervised? What are the underlying algorithms? How does the underlying neural network learn from the non-monotonic spiking output of TSM? In second paragraph, page 9, it is written that, “In case of straight moving, as the robot car becomes close to the light source, the spiking frequency of LGMD neuron increases and robot car at a constant speed will turn an angle to avoid the looming collision”. How is the non-monotonicity of spiking output from TSM used for collision avoidance if the car turns just with increasing spiking frequency? It is strongly encouraged to provide a detailed explanation of Fig. i, j and supplementary figures 13, 14 and the underlying algorithms of collision detection.

ANS: We are sorry that we should not define the demonstration of collision avoidance as unsupervised spike neural network. As the reviewer mentioned about the spiking output from TSM used for collision avoidance, the light-shifted neuron firing probabilities in Fig. 6f exhibits a non-monotonicity evolution with the input light intensity. When the UV light irradiance increases from 0.00 mW to 0.37 mW, the firing probability increases from 0.032,

0.044, 0.052 to 0.080. However, a much higher irradiance of 2.50 mW inhibits the firing response of the LGMD neuron and reduces the firing probability to 0.052. The output firing spike frequency controls the turning speed of the robot car, Supplementary movie 1-3 evidenced the approaching process of the robot car to the forward light source. In case of straight moving, as the robot car becomes close to the light source, the spiking frequency of LGMD neuron increases and robot car at a constant speed will turn an angle to avoid the looming collision (irradiance of 1mW, corresponding to the peak response of LGMD neuron). On the contrary, the robot car will move linearly when the object is far from it (irradiance of 0.5 mW and 2.5 mW, corresponding to the excitatory and inhibitory response of LGMD neuron, respectively). The LGMD neuron can derive a directional decision from different optical inputs according to f_{spike} .

We provide the detailed explanation of the revised Fig. 6i, j and Supplementary Figures 13, 14 (revised as Supplementary Fig. 31, 32) and the underlying operation principle of collision detection in the manuscript as below.

In the manuscript:

“To demonstrate the potential integration of artificial LGMD neurons with circuits, the f_{spike} controlled system is proposed to accomplish the collision avoidance function of the robot car. The decision-making process for the robot car in the motion trajectory is explained in Fig. 6i, j and Supplementary Fig. 31, 32. In addition to Fig. 6k, Supplementary Movies 1-3 also demonstrate the process of the robot car approaching the imminent stimulus. According to the characteristics of the LGMD neuron model in Fig. 6h, its avoidance behavior should appear at the highest point, which is defined as 2.5 Hz. By using a counter to count the fire pulse number within 10 s to obtain f_{spike} compared with 2.5 Hz, microcontroller can be driven to achieve the output of 0 or 1. As the robot car approaches the light source, the f_{spike} of the LGMD neuron increases. When f_{spike} reaches 2.5 Hz, the robot car will turn an angle to avoid imminent collision. Conversely, the robot car will move linearly with a f_{spike} less than 2.5 Hz. Thus, the output f_{spike} controls the turning time of the robot car, which derives a directional decision for the LGMD neuron under different situations. This proof-of-principle demonstration of the fully integrated TSM neuron can be expanded to implement learning systems of higher complexity in an energy-efficient manner.”

In Supporting Information:

“The circuit design mainly includes the following parts:

Zone A named power supplement part: the 12 V input voltage is converted to 5 V by the 78L05 chip, the 5 V voltage is converted to 3.3 V by the SC662K chip. Among them D1 is the power indicator light.

Zone B named controller part: STM32 and its peripheral circuit control the corresponding output pulse waveform and control the drive circuit according to the input voltage signal.

Zone C named serial port part: download the program and monitor the running status of the program

Zone D named amplifier part: convert the controller output voltage of 3.3 V to 1 V.

Zone E named driving circuit: receive the signal from the controller and controls the car to go straight or turn.

Zone F named light sensor circuit: The pulse signal output by the controller is passed through memristor to obtain the output voltage spiking frequency, and the output voltage spiking frequency is compared with the reference spiking frequency through a comparator. Finally, the output voltage spiking frequency is transmitted to the controller. If the output voltage spiking frequency of the comparator is high than 2.5 Hz, the controller controls the car to turn.

Working principle: The Zone B controller outputs a 3.3 V pulse waveform and transmits it to the Zone D amplifier, and the amplifier converts the pulse into 1V pulse and inputs it to the Zone F light sensor circuit. The light sensor circuit uses the output voltage obtained by the comparator to determine whether it is fire by comparing the relationship between the output voltage spiking frequency and 2.5 Hz, and counts the number of firing through the internal counter of the single-chip microcomputer to determine whether it reaches the avoidance frequency. Finally, the result is sent back to the control circuit, and the control circuit controls the driving circuit to go straight or turn according to the Zone E.”

4) Having a flexible photosensitive array will certainly help in creating hemispherical vision system, which allows achieving large FoV artificial vision systems. But a concrete integration of a large FoV artificial vision system with its previous section consisting of collision avoidance is missing. The reviewer believes that putting more emphasis on this aspect and associated demonstrations can significantly improve the quality of the manuscript.

ANS: To investigate the reliability of the large FoV artificial vision system, we performed the collision avoidance investigation on the flexible TSM array. With the fixed light irradiation, 9 special positioned devices with incident angles of 10°, 20°, 30°, 40°, 50°, 60°, 70°, 80° and 90° were selected for collision avoidance investigation (Supplementary Fig. 23, 24).

We added the relevant information to the manuscript and the Supporting Information as below.

In manuscript:

“To investigate the production yield of the FLBP-CsPbBr₃ TSM device-based crossbar array, we measured all the 400 devices in a 20×20 crossbar array. The *I-V* characteristics of the 400 memristor devices are provided in Supplementary Fig. 15-18, with 337 of them showing stable threshold switching characteristics and a production yield of 84.25%.

... The operation principle of our artificial LGMD neuron is depicted in the Fig. 4f, h. Notably, an early switching event will be triggered at a higher temperature (Supplementary Fig. 21), which is also in consistent with the second-order memristor effect.

Concrete integration of a large FoV artificial vision system with the non-monotonic response of TSM device were demonstrated by measuring the current values of FLBP-CsPbBr₃ TSM on the hemispherical visual system in specific position from 0° to 90° under a series of electrical pulses in response to different laser ramp rates. Supplementary Fig.

23 illustrates that 9 specific positioned TSM device were selected with the light incident angle from 0° to 90° . During the measurements, a voltage stimuli train consisting of $+0.2$ V pulses, duration of $50 \mu\text{s}$ and interval of $50 \mu\text{s}$ were employed. The ramp rate of laser light can be used to capture different looming speeds. The laser intensity was ramped from 0.00 mW to 2.50 mW with different time intervals to determine the increased intensity rate in each case. Obviously, the response of current is non-monotonic for all the measurements (see the illustration of Fig. 5a and Supplementary Fig. 24). Fig. 5b maps the inflection point values of TSM output current as per the data shown in Fig. 5a. In addition, 3D spatial distribution of the time-to-collision detection (τ_D) of our large FoV artificial vision system is demonstrated in Fig. 5c. The further optimizations of our flexible artificial vision system with incident light may potentially bring forth applications in the vision chip of autonomous robots for indicating collision.”

In Supporting Information:

“

Supplementary Figure 15. *I-V* characteristics of the first 100 FLBP-CsPbBr₃ TSM devices in a 20×20 crossbar array. All the *I-V* curves were obtained by sweeping the voltage in the sequence of $0 \text{ V} \rightarrow 3 \text{ V} \rightarrow 0 \text{ V}$.

Supplementary Figure 16. *I-V* characteristics of the second 100 FLBP-CsPbBr₃ TSM devices in a 20 × 20 crossbar array. All the *I-V* curves were obtained by sweeping the voltage in the sequence of 0 V → 3 V → 0 V.

Supplementary Figure 17. *I-V* characteristics of the third 100 FLBP-CsPbBr₃ TSM devices in a 20 × 20 crossbar array. All the *I-V* curves were obtained by sweeping the voltage in the sequence of 0 V → 3 V → 0 V.

Supplementary Figure 18. *I-V* characteristics of the fourth 100 FLBP-CsPbBr₃ TSM devices in a 20 × 20 crossbar array. All the *I-V* curves were obtained by sweeping the voltage in the sequence of 0 V → 3 V → 0 V.

Supplementary Figure 23. Specific 9 positions with incident angles of 10°, 20°, 30°, 40°, 50°, 60°, 70°, 80° and 90° for the subsequent collision avoidance investigation.

Supplementary Figure 24. Output current of the different positioned FLBP-CsPbBr₃ TSM

device in response to different looming object speed.”

5) As mentioned in the previous comment, the hemispherical artificial vision system is the most interesting aspect of this manuscript. In the section titled “Biomimetic compound eye based on artificial LGMD neuron”, after explaining benefits of having a hemispherical artificial vision system, the wavelength-dependency is discussed. The reviewer struggles to understand how this paragraph can be part of this section.

ANS: Emulating the biological visual perception system typically requires a complex architecture including the integration of an artificial retina and optic nerves with various synaptic behaviors. In the biological visual systems, the color-selective imaging process enriches the visual perception. As shown in Fig. R1, the color graph helps to distinguish between the plants and the flowerpot. The efficient light management in the natural eye is an important natural facet that should be mimicked in artificial eyes.

Recently, various optoelectronic neuromorphic devices have been investigated for developing artificial visual perception system. The conversion of light signals into electrical signal with wavelength-discrimination have been one of the challenges for the artificial visual perception system. In order to vividly mimic the biological visual system, optoelectronic devices were explored to demonstrate color-selective imaging process and neuromorphic functions. [*Adv. Mater.*, **2018**, 30, 1803961; *Adv. Mater.*, **2019**, 31, 1806227; *Adv. Mater.*, **2019**, 31, 1906433; *Sci. Adv.*, **2018**, 4, eaat7387.]

The two-terminal memristor device offer a simple device structure which can be simply integrated into the electronic circuit to emulate the artificial eyes, but it also has limitations such as low capability of wavelength discrimination. Thus, most reported memristor based artificial vision systems exhibited narrow-band imaging capability. Here, the *I-V* characteristics of FLBP-CspbBr₃ TSM measured under positive sweep with different light wavelengths in the manuscript illustrate that our TSM device based artificial vision system can be designed for a relatively broad band discrimination from 365 nm to 520 nm. This artificial vision system with wavelength-dependency property ensures the emulation of biological color-selective imaging process.

Figure R1. Representative image of a cactus in a blue and purple basin.

6) A thorough explanation of Figure g and i will be helpful. Different wavelength radiation will invoke different threshold shifts. Also, for a given wavelength radiation, different devices at different angles in the hemispherical array will respond differently. How can the system differentiate between wavelength dependent response and angle-dependent response?

ANS: As the wavelength dependent response and angle-dependent response are interactive with the photocurrent signal of the TSM device, we use the MATLAB software to fit the formula between the output current, light wavelength and incident angle based on least-squares method. As per the reviewer's suggestion, we added the relevant information to the Supporting Information as below.

In Supporting Information:

“Supplementary Fig. 22 shows the output current as a function of light wavelength and the incident angles. By using the least-squares method with the MATLAB software, the desired output current $f(x,y)$ for the TSM device is determined by

$$f(x,y)=a_1+a_2x+a_3y+a_4x^2+a_5xy+a_6y^2+a_7x^3+a_8x^2y+a_9xy^2+a_{10}x^4+a_{11}x^3y+a_{12}x^2y^2+a_{13}x^5+a_{14}x^4y+a_{15}x^3y^2$$

where x is angle, y is wavelength, $a_1=0.2444$ (0.1635, 0.3254), $a_2=0.07342$ (-0.1191, 0.266), $a_3=-0.3349$ (-0.392, -0.2777), $a_4=-0.2742$ (-0.4382, -0.1102), $a_5=0.01547$ (-0.06, 0.09095), $a_6=0.1546$ (0.09387, 0.2154), $a_7=-0.1272$ (-0.4194, 0.165), $a_8=0.2437$ (0.103, 0.3844), $a_9=-0.01567$ (-0.1228, 0.0915), $a_{10}=0.07682$ (0.01436, 0.1393), $a_{11}=-0.003562$ (-0.04078, 0.03365), $a_{12}=-0.06462$ (-0.1099, -0.01931), $a_{13}=0.04099$ (-0.05244, 0.1344), $a_{14}=-0.04444$ (-0.09662, 0.007733), $a_{15}=0.002848$ (-0.04929, 0.05498).

The light wavelength and incident light can be well decoupled according to the above function.

Supplementary Figure 22. The current as a function of light wavelength and the incident angle along x direction (a) and y direction (b).”

7) “Apart from the light intensity sensing behavior demonstrated in Fig. 6b, the wavelength-dependent recognition improves the classification accuracy of various objects” – What kind of classification accuracy are authors referring to?

ANS: Both the light intensity and the light wavelength play a dominant role in defining the quality of imaging for the biological visual perception system. As described by Prof. Burn, “high level of color purity is usually required for applications such as real-time video surveillance that demand greater accuracy in imaging objects in natural scenes” (*Adv. Mater.*, **2016**, 28, 4766–4802), thus both light sensitivity and color-selectivity are the essential characteristics for the artificial vision system.

Since the FLBP-CsPbBr₃ TSM can generate various levels of photocurrent corresponding to irradiated input-light spectrum, distinctive wavelength-dependent response determines the spectral sensitivity and color accuracy. In order to give a better understanding, we correct sequence of Fig.6 as Fig.4 with the revised the sentence in the manuscript as below.

In the manuscript:

“Apart from the light **power** sensing behavior demonstrated in Fig. 3b, the **wavelength-dependent effect of threshold switching characteristics improves the essential light sensitivity and color-selectivity for the artificial vision system, which would improve the perception accuracy of the system.**”

8) What is V_F and how is it determined?

ANS: “ V_{spike} ” stands for “spiking frequency”. “ V_F ” stands for “firing frequency”, which is defined as 2.5 Hz in our manuscript. The defined value is based on the collision detecting spiking frequency in Fig. 6h. In order to give a better understanding, we corrected the “ V_{spike} ” as “ f_{spike} ”, and term “ V_F ” as 2.5 Hz in the manuscript as below.

In the manuscript:

“As shown in Fig. 6f, the spiking frequency (f_{spike}) strongly depends on the power of light. According to the previous results, the conductance of FLBP-CsPbBr₃ increases, peaks and decreases as the power of UV illumination increases (Fig. 6h). The f_{spike} follows the similar trend as the increase of UV irradiance which is analogue to the response of LGMD neuron to a looming object (Fig. 6g), and reaches to a 2.5 Hz (25 fired pulse within 10 s) to detect the subsequent collision. Whereas the influence of light wavelength on the f_{spike} is demonstrated in Supplementary Fig. 30. The f_{spike} decreases as the light wavelength shift from 365 nm to 520 nm.”

9) Natural compound eyes like apposition and refracting/reflecting superposition compound eyes comes with corneal lens to focus light onto the rhabdom. Is it possible to integrate optical lenses with the FLBP-CsPbBr₃ TSM flexible array?

ANS: As the reviewer mentioned, natural compound eyes come with corneal lens to focus light onto the rhabdom. Man-made biomimetic systems can be grouped according to the biological designs of the compound eyes. Herein, we integrated the optical lens with FLBP-CsPbBr₃ TSM flexible array with Fig. R2 below. The incident UV light is focused onto the FLBP-CsPbBr₃ TSM flexible array. Besides, three alternative approaches can also be utilized to integrate optical lenses with the flexible device array, which are listed as below:

(1) Design a shell to insert the device array and the optical lens. In fact, Prof. Fan presents an electrochemical eye with a hemispherical retina, as illustrated in the Fig. R3a below (*Nature*, **2020**, 581, 278-282). A convex lens was then glued to front shell to complete the device fabrication.

(2) Scaling down the size of the optical lens to microlenses to integrate into the artificial compound eyes system. As shown in Fig. R3b (*Nature*, **2013**, 497, 95-99), an array of elastomeric microlenses are assembled onto the hemispherical, apposition compound eye camera.

(3) Design metasurface to allow a lensless compound-eye vision. Recently, Professor Paiella describes a compound-eye camera based on a fundamentally different approach to mimic the compound-eye vision. (*Nature Communications*, **2020**, 11, 1637) The integration of each pixel of a standard image-sensor array with a specially designed metasurface in Fig. R3c that only allows for the detection of light incident along a small, geometrically tunable distribution of angles, whereas light incident along all other directions is reflected. Based on metallic plasmonic nanostructures combined with simple Ge photoconductors, ultrathin planar cameras can be developed without any lenses.

Figure R2. a. Representative image of the device test system with an optical lens. b. Schematic illustration of the lens integrated FLBP-CsPbBr₃ TSM flexible array.

Editorial Note: panels a & b below reprinted respectively from Gu, L. *et al.* A biomimetic eye with a hemispherical perovskite nanowire array retina. *Nature* **581**, 278-282 (2020) and Song, Y. M. *et al.* Digital cameras with designs inspired by the arthropod eye. *Nature* **497**, 95-99 (2013), with permission from Springer Nature.

Figure R3. a. Detailed structure of the electrochemical eye with a hemispherical retina (*Nature*, 2020, 581, 278-282). b. Illustration of the elastomeric microlenses assembled position compound eye camera (*Nature*, 2013, 497, 95-99). c. Compound-eye camera based

on the angle-sensitive metasurfaces developed in the present work, where only light incident along a different direction is transmitted into each image sensor (*Nature Communications*, **2020**, 11, 1637).

10) Authors are encouraged to reduce spelling mistakes in the manuscript. For example 'TMS' in Figure 6e, and 'LGMD respnse' in Figure 5g. These are not critical and not deciding factors for considering the manuscript for publication, but still can provide a better impression for the reader.

ANS: Thanks for your correction! We have revised the mistakes in the revised manuscript.

Reviewer #2 (Remarks to the Author):

In this paper, the authors reported an lobula giant movement detector (LGMD) neuron based on a light-mediated threshold switching memristor. The light-dependent escape response of the LGMD neuron and its mechanisms were systematically demonstrated. Furthermore, the light-dependent properties of the LGMD neurons were implemented in robot navigation with obstacle avoidance and artificial eyes with wide detection functions. In my view, this work is interesting but some issues should be solved for further improvement.

ANS: Thank you for reviewing our paper. We appreciate your insightful comments on our research. We have revised the manuscript according to your suggestions and believe that these revisions have improved the paper.

Please find below our responses (in blue) to each of your specific comments (in black). Revisions to the original article are indicated in red.

1. The authors used BP and halide perovskite of CsPbBr₃ quantum dots to fabricate the two-terminal memristors. Why did the authors specifically used BP and CsPbBr₃? Did the authors consider the band alignment of the BP and CsPbBr₃ in material selection? Also, the authors should explain the electrons and holes transport under the light illumination and the effect of generated electrons and holes in terms of device operation and performance.

ANS: In order to characterize the photoresponse behavior under a constant light source and explain the underlying mechanism, we evaluated the surface potential via the KPFM measurements. Light irradiance with different intensities (0.00 mW, 0.24 mW, 0.72 mW and 2.50 mW) was applied to the FLBP-CsPbBr₃ layer. We added the relevant information to the manuscript and the Supporting Information as below.

In manuscript:

“The underlying mechanism of light sensing capability of FLBP-CsPbBr₃ TSM were investigated by the Kelvin probe force microscopy (KPFM) measurements. The surface potential characteristics were evaluated under a UV illumination. For comparison, the light irradiance with different intensities was applied to the FLBP-CsPbBr₃ layer. As displayed in the Supplementary Figure 11, the surface potential steadily increases from 69.2 mV, 134.0 mV, 211.7 mV to 273.2 mV when the light power increases from 0.00 mW, 0.24 mW, 0.72 mW to 2.50 mW. Significant surface potential enhancement was observed in the FLBP-CsPbBr₃ layer since the light illumination on the CsPbBr₃ QDs film can trigger generation of photocarriers. The photogenerated electrons can be easily transferred from the CsPbBr₃ to FLBP through the internal electric field and thus photoinduced holes are left in the valence band of the CsPbBr₃ which induces the increase of surface potential.”

In Supporting Information:

“As shown in the Supplementary Fig. 11, the conduction band (valence band) of the few layered FLBP and CsPbBr₃ QDs is -4.58 eV (-4.67 eV) and -3.3 eV (-5.7 eV), respectively, hence the hybrid junction is expected to be type-I with a band offset of 1.28 eV (The electronic band structure of the FLBP varies with the layer numbers. Here we take 5 layered FLBP as an example). The large band offset induces the separation of the photogenerated electron-hole pairs in CsPbBr₃ and transportation of the electrons to FLBP. Additionally, the photoinduced electron transfer from CsPbBr₃ to FLBP was further confirmed by the KPFM measurements in Supplementary Fig. 12.

Supplementary Figure 11. Sketch of the conduction and valence band profiles and the electron-hole dynamics under irradiation.

Supplementary Figure 12. Surface potential of the FLBP-CsPbBr₃ layer under different light illumination (fixed light wavelength: 365 nm) recorded by in situ AFM electrical nano-technology (scale bar, 100 nm). The lower panel shows the respective surface potential profile.”

2. For automotive applications, the device should show stable and consistent characteristics under high temperatures. However, as the operation mechanism of the suggested FLBP-CsPbBr₃ TSM is based on Joule heating, the temperature of the device might highly affect the switching characteristics of the device. The authors should provide the electrical

characteristics under diverse temperatures to ensure their applicability.

ANS: To evaluate the temperature effect on the performance of the FLBP-CsPbBr₃ TSM, the threshold switching characteristics of device were measured at different temperatures. We provide the electrical characteristics of the device under diverse temperatures in the manuscript and Supporting Information as below.

In the manuscript:

“The operation principle of our artificial LGMD neuron is depicted in the Fig. 3f, h. Notably, an early switching event will be triggered at a higher temperature (Supplementary Fig. 21), which is also in consistent with the second-order memristor effect.”

In Supporting Information:

“The substantial output signals shown in Supplementary Fig. 21 suggests the onset of a switching event. It was discovered that at a higher temperature, the early switching event will be triggered. When the temperature was increased from 298 K to 373 K, the amplitude of the current signal increased while the switching time of the signal decreased.

Supplementary Fig. 21 plot temperature dependent characteristics of devices by applying fixed 0.2 V voltage pulses (50 μs in width, 50 μs in interval). The current increases with temperature in the pre-switching state, suggesting a thermally activated electron hopping mechanism¹. After occurrence of switching event, the current value decreases as the temperature increases, suggesting that the conduction mechanism follows a metallic behavior. This result implies that there would be residual Ag clusters existed between Ag electrode and ITO electrode to reduce the effective gap distance, which can effectively engineer the conduction following a thermal-assistant hopping mechanism.

Supplementary Figure 21. Excitatory and inhibitory response of the device to a looming light stimulus with simultaneously applied programming electronic pulses under diverse temperatures ranging from 298 K to 372 K (0.2 V voltage pulse, 50 μ s duration, 50 μ s interval).”

3. Another concern is the power consumption of the suggested device. To achieve obstacle avoidance, voltage pulses with long duration (50 ms) and interval (50 ms) should be consistently applied, which will lead to considerably high power consumption compared to previous reports on LGMD neurons (Nature Electronics 3, 646-655 (2020)).

ANS: As per the reviewer’s suggestion, we roughly calculated a writing energy consumption ($P_{writing}$) for a single operation using the equation as below:

$$P_{writing} = V_{amp} \times I_{peak} \times t$$

V_{amp} and I_{peak} represent the amplitude of the applied writing voltage and the peak value of the writing current, respectively. t is the pulse width of applied writing voltage. The energy consumption was estimated as $P_{writing} = 2.03 \times 10^{-6} \text{ A} \times 0.2 \text{ V} \times 0.05 \text{ s} = 20.3 \text{ nJ}$ for the 50 ms width pulse. When we use the voltage pulse with duration of 50 μ s, interval of 50 μ s and amplitude of 1.2 V (experimental parameters for collision avoidance tests), the read-out current increases to 0.035 mA with the voltage stimulus, where the power consumption was estimated as $P_{writing} = (3.5 \times 10^{-5} \text{ A}) \times (1.2 \text{ V}) \times (5 \times 10^{-5} \text{ s}) = 2.10 \times 10^{-9} \text{ J}$ (Supplementary Fig. 29).

As shown in Fig. R4a, with the applied V_{BG} of -0.25 V, the read-out current in previous reports on LGMD neurons (*Nature Electronics*, **2020**, 3, 646-655) is about $\sim 7.5 \times 10^{-10}$ A/ μm^2 with the programming pulse amplitude of 8.5 V, pulse width of 100 ms, and device area of $5 \mu\text{m}^2$, the power consumption is thus calculated to be $\sim 7.5 \times 10^{-10}$ A/ $\mu\text{m}^2 \times 5 \mu\text{m}^2 \times 8.5 \text{ V} \times 0.1 \text{ s} = 3.19 \times 10^{-9}$ J. By changing the V_{BG} to -1.0 V (Fig. R4b), the power consumption is reduced to be $\sim 1.0 \times 10^{-12}$ A/ $\mu\text{m}^2 \times 5 \mu\text{m}^2 \times 8.5 \text{ V} \times 0.1 \text{ s} = 4.25 \times 10^{-12}$ J. Fig. R4c shows a bar plot for the time-to-collision detection. The pink bars represent the time to collision for different object speeds from which the prediction of the collision time is more accurate with the applied V_{BG} of -0.25 V. The trade-off between the low energy and the prediction accuracy should be taken into consideration when different V_{BG} is employed.

Shortening the programming time is an effective way to reduce $P_{writing}$. When we scale the pulse width down to 50 μs , the writing energies for the applied pulses decreased to 2.10×10^{-9} J. Comparing to the previous reports on LGMD neurons (*Nature Electronics*, **2020**, 3, 646-655) with the applied V_{BG} of -0.25 V, our TSM device can achieve a considerably lower power consumption by reducing the operation time. Meanwhile, the higher read-out current of our TSM device made a more accurate detection on the approaching object. We calculated the writing energy of the device in the manuscript and added the discussion about the scaling of writing energies to the Supporting Information as below.

In the manuscript:

“Additionally, we investigated the writing energy of the device for single voltage pulse by measuring the current between two terminals (Supplementary Fig. 29). The device exhibited energy consumption of 20.3 nJ for the 50 ms width pulse, which was further reduced to 2.1 nJ under a pulse width of 50 μs .”

In Supporting Information:

“

Supplementary Figure 29. Output currents measured in response to the programming pulse trains with an interval of 50 μs and width of 50 μs .”

Editorial Note: Figure R4 reprinted from Jayachandran, D. *et al.* A low-power biomimetic collision detector based on an in-memory molybdenum disulfide photodetector. *Nat. Electron.* **3**, 646-655 (2020) with permission from Springer Nature.

Figure R4. Output currents and measured time to collision under different ‘read’ conditions in response to the ‘write’ programming pulse trains of different amplitudes.

4. In Figure 2b, the memristors show the threshold switching behavior with threshold voltage of about 1 V during DC voltage sweep. However, in Figure 2f, when a voltage pulse (0.6 V, 20 ms) is applied, the device showed threshold switching behavior. How is it possible to change the resistance of the device at a pulse amplitude smaller than the threshold voltage observed in DC voltage sweep?

ANS: The reviewer mentioned about the threshold voltage of 0.6 V in Fig. 2f, which was lower than the V_{th} in I - V sweeps (Fig. 2b). This is because longer programming time can slightly reduce the switching voltage of memristors. For the following measurements, we applied a train of 0.6 V voltage pulses with variable durations (2 ms to 50 ms) to investigate the relationship between the pulse duration and the response currents. As shown in the Supplementary Fig. 6 (see below), the current increases steadily with the increased pulse duration. The similar trend was observed in the other threshold switching memristor, which is illustrated in the Fig. R5 as below (*Nat. Commun.*, **2021**, 12, 3351).

We added the relevant information to the manuscript and Supporting Information as below.

In the manuscript:

“It is worth pointing out that a long interval time of 0.15 s was employed in between these single pulse measurements to ensure that the TSM has enough time to decay back to its original HRS^{41,42}. Here the switching voltage of 0.6 V was lower than the V_{th} in Fig. 2a, because the programming operation with relatively longer time can induce resistance transition with a lower voltage (Supplementary Fig. 6).”

In Supporting Information:

“

Supplementary Figure 6. A train of 0.6 V applied voltage pulses of variable durations (2 ms to 50 ms) (top panel) and response currents (bottom panel). A longer input pulse triggers a higher response current.”

Figure R5. Demonstration of the switching I - V curves and the potential evolution in the memristor. (*Nature Communications*, 2021, 12, 3351)

5. The authors claimed that the voltage pulse with higher amplitude and longer duration leads to longer relaxation time with a narrower distribution (Figure 2g). Although the relaxation time increases with increasing amplitude and duration, the distribution seems to be irrelevant with the condition of voltage pulse.

ANS: As per the reviewer's suggestion, we revised the description in the manuscript. The current value of the TSM device would abruptly jump to several orders of magnitude under an applied pulse and relaxed back to its original state over a characteristic time τ . Thus, we calculated the variability of the relaxation time for each condition of the voltage pulse to demonstrate the spread of the device performance, which is defined based on the standard deviation and the means of the relaxation distributions. As shown in Table R1, the variability of the relaxation time for each condition of the voltage pulse is small enough to verify the narrow distribution for all the relaxation process.

Table R1. The variability of the relaxation time for each condition of the voltage pulse.

Pulse width (ms)	20	20	20	20	20	10	50
Pulse amplitude (V)	0.6	0.8	1.0	1.2	1.4	1.0	1.0
Variability	0.205	0.120	0.085	0.044	0.038	0.215	0.081

We modified the manuscript as below.

In the manuscript:

“The voltage with higher amplitude and longer duration leads to the longer relaxation time with a narrow distribution.”

6. In Figure S7, the authors claimed that the electrical properties of the device were changed depending on the concentration of CsPbBr₃. However, the explanation of the characteristics is insufficient. Also, in inset images, the thin film characteristics such as grain size, film quality, and roughness seem to be different. Did these film characteristics affect the electrical properties? In addition, information on the film thicknesses of each concentration should be presented for accurate comparison.

ANS: The average roughness of FLBP-CsPbBr₃ layer varies with the concentration of CsPbBr₃. As shown in Supplementary Fig. 13, the FLBP-CsPbBr₃ (50%) and FLBP-CsPbBr₃ (75%) exhibit roughness of 2.63 nm and 2.83 nm, respectively, suggesting the good quality of the FLBP-CsPbBr₃ layer. Relative higher proportion of FLBP in the FLBP-CsPbBr₃ (50%) exhibit higher film conductivity. In addition, the film thicknesses of the samples with different concentration of CsPbBr₃ are compared in Supplementary Fig. 14 as below. The film thickness of the FLBP-CsPbBr₃ (50%) is thinner than FLBP-CsPbBr₃ (75%). Thus, a smaller on/off ratio of FLBP-CsPbBr₃ (50%) based device is obtained in the *I-V* characterization in comparison with the FLBP-CsPbBr₃ (75%) based device.

The grain size of the resistive switching material will affect the ion migration behavior in the device and result in different memristive performance. As reported by Prof. Huang, film with large grain size make the migration easier and result in a better performance (*Phys. Chem. Chem. Phys.*, **2016**, 18, 30484-30490). The activation energies for different grain size of MAPbI₃ films is about 0.27 eV (~0.3 μm), 0.50 eV (~1 μm) and 1.05 eV (single crystal, represents the extreme case of large grain film). The activation energies of ion migration in single crystals are at least two-fold larger than those of the polycrystalline films. Herein, we estimate the grain size of the assembled CsPbBr₃ QDs in Supplementary Fig. 14 as below. The average grain size of CsPbBr₃ QDs is 8.594±0.152 nm (FLBP-CsPbBr₃(50%)), 8.846±0.099 nm (FLBP-CsPbBr₃(66%)), 9.697±0.195 nm (FLBP-CsPbBr₃(75%)) and 11.347±0.099 nm (FLBP-CsPbBr₃(86%)), which are much smaller than the above mentioned MAPbI₃ films with grain size of ~1 μm. Thus, the negligible variation of grain size in our work may not have a significant influence on the memristive performance. We added the discussion about the film roughness, grain size and the film thickness of FLBP-CsPbBr₃ in the Supporting Information as below.

In Supporting Information:

“The average roughness of FLBP-CsPbBr₃ layer varies with the concentration of CsPbBr₃. As shown in Supplementary Fig. 13, the FLBP-CsPbBr₃(50%) and FLBP-CsPbBr₃(75%) exhibit roughness of 2.63 nm and 2.83 nm, respectively, suggesting the better quality of the FLBP-CsPbBr₃ layer. Relative higher proportion of FLBP in the FLBP-CsPbBr₃ (50%) exhibit higher film conductivity. In addition, the film thicknesses of the samples with

different concentration of CsPbBr₃ are compared in Supplementary Fig. 14. The film thickness of the FLBP-CsPbBr₃ (50%) is thinner than FLBP-CsPbBr₃ (75%). Thus, a smaller on/off ratio of FLBP-CsPbBr₃ (50%) based device is obtained compared with the FLBP-CsPbBr₃ (75%) based device.

Supplementary Figure 13. Demonstration of the roughness of the different FLBP-CsPbBr₃ film and the grain size of assembled CsPbBr₃ QDs.

Supplementary Figure 14. Cross-sectional SEM image of the vertical stack of the TSM fabricated with FLBP-CsPbBr₃(50%), FLBP-CsPbBr₃(66%), FLBP-CsPbBr₃(75%) and FLBP-CsPbBr₃(86%).”

7. The statement “The FLBP-CsPbBr₃ TSM initially exhibited a resistance of $6.25 \times 10^{-11} \Omega$ at HRS” on page 2 seems to be wrong. The current of TSM at HRS is about 10^{-11} A under 0.1 V, which leads to resistance over 10^{10} .

ANS: As per the reviewer’s suggestion, we revised statement about the resistance value. With the reading voltage of 0.1 V, the HRS current for the red curve in Figure 2a is 6.25×10^{-11} A. Therefore, the resistance at HRS should be $1.6 \times 10^9 \Omega$. We corrected it in the manuscript as below.

In the manuscript:

“The FLBP-CsPbBr₃ TSM initially exhibited a resistance of $1.6 \times 10^9 \Omega$ at high-resistance state (HRS) with the reading bias of 0.1 V (Fig. 2a).”

8. The authors demonstrate the 20×20 TSM flexible array on the PDMS substrate for biomimetic compound eye. The authors should provide additional data to show the reliability of the array in a flexible environment.

ANS: To investigate the flexibility of the device, we performed the bending tests on the device with different bending angles. We added the corresponding information to the Supporting Information as below.

In Supporting Information:

“The corresponding I - V characterization with varied bending angles was provided in Supplementary Fig. 5. The stable threshold switching performance as the device array is bending from 9° to 320° demonstrates the feasibility of the flexible applications.

Supplementary Figure 5. Typical I - V curves with respect to the different curvatures of the TSM. The optical images of memristor with different bending angles are shown in the upper panel.”

Furthermore, to investigate the reliability of the large FoV artificial vision system, we performed the collision avoidance investigation on the flexible TSM array. With the fixed light irradiation, 9 special positioned devices with incident angles of 10° , 20° , 30° , 40° , 50° , 60° , 70° , 80° and 90° were selected for collision avoidance investigation (Supplementary Fig. 24).

We added the relevant information to the manuscript and the Supporting Information as below.

In manuscript:

“To investigate the production yield of the FLBP-CsPbBr₃ TSM device-based crossbar array, we measured all the 400 devices in a 20×20 crossbar array. The I - V characteristics of

the 400 memristor devices are provided in Supplementary Fig. 15-18, with 337 of them showing stable threshold switching characteristics and a production yield of 84.25%.

... The operation principle of our artificial LGMD neuron is depicted in the Fig. 4f, h. Notably, an early switching event will be triggered at a higher temperature (Supplementary Fig. 21), which is also in consistent with the second-order memristor effect.

Concrete integration of a large FoV artificial vision system with the non-monotonic response of TSM device were demonstrated by measuring the current values of FLBP-CsPbBr₃ TSM on the hemispherical visual system in specific position from 0° to 90° under a series of electrical pulses in response to different laser ramp rates. Supplementary Fig. 23 illustrates that 9 specific positioned TSM device were selected with the light incident angle from 0° to 90°. During the measurements, a voltage stimuli train consisting of +0.2 V pulses, duration of 50 μs and interval of 50 μs were employed. The ramp rate of laser light can be used to capture different looming speeds. The laser intensity was ramped from 0.00 mW to 2.50 mW with different time intervals to determine the increased intensity rate in each case. Obviously, the response of current is non-monotonic for all the measurements (see the illustration of Fig. 5a and Supplementary Fig. 24). Fig. 5b maps the inflection point values of TSM output current as per the data shown in Fig. 5a. In addition, 3D spatial distribution of the time-to-collision detection (τ_D) of our large FoV artificial vision system is demonstrated in Fig. 5c. The further optimizations of our flexible artificial vision system with incident light may potentially bring forth applications in the vision chip of autonomous robots for indicating collision.”

In Supporting Information:

“

Supplementary Figure 15. *I-V* characteristics of the first 100 FLBP-CsPbBr₃ TSM devices in a 20×20 crossbar array. All the *I-V* curves were obtained by sweeping the voltage in the sequence of 0 V → 3 V → 0 V.

Supplementary Figure 16. *I-V* characteristics of the second 100 FLBP-CsPbBr₃ TSM devices in a 20 × 20 crossbar array. All the *I-V* curves were obtained by sweeping the voltage in the sequence of 0 V → 3 V → 0 V.

Supplementary Figure 17. *I-V* characteristics of the third 100 FLBP-CsPbBr₃ TSM devices in a 20 × 20 crossbar array. All the *I-V* curves were obtained by sweeping the voltage in the sequence of 0 V → 3 V → 0 V.

Supplementary Figure 18. *I-V* characteristics of the fourth 100 FLBP-CsPbBr₃ TSM devices in a 20 × 20 crossbar array. All the *I-V* curves were obtained by sweeping the voltage in the sequence of 0 V → 3 V → 0 V.

Supplementary Figure 23. Specific 9 positions with incident angles of 10°, 20°, 30°, 40°, 50°, 60°, 70°, 80° and 90° for the subsequent collision avoidance investigation.

Supplementary Figure 24. Output current of the different positioned FLBP-CsPbBr₃TSM

device in response to different looming object speed.”

Reviewer #3 (Remarks to the Author):

The authors present a novel study ranging from threshold switching memristor to artificial visual neuron. The proposed non-monotonic response of the device to light flow field could achieve bioinspired escape response with the implemented artificial neuron circuit, allowing an obstacle avoidance capability in robot navigation. The authors systematically investigated the influence of different looming object speed and voltage stimulus on output conductance so as to achieve the optimal collision detection. Additional, biomimetic compound eye with hemispherical retina made of flexible device arrays showed the applicability of wide field-of-view detection. This original paper is clear in its theme, profound in its experimental and theoretical research. The topic will inspire further research in this field. Therefore, the reviewer recommends that the paper is accepted with major revisions on addressing points outlined below. These comments are not to criticize but to enhance the manuscript for publishing in high-impact Nature Communications.

ANS: Thank you for reviewing our paper. We appreciate your insightful comments on our research. We have revised the manuscript according to your suggestions and believe that these revisions have improved the paper.

Please find below our responses (in blue) to each of your specific comments (in black). Revisions to the original article are indicated in red.

1. Fixing device variability is a prerequisite for mass production. What about the device-to-device variability?

ANS: To demonstrate the device-to-device variability, we have done additional experiments to discuss the production yield of the FLBP-CsPbBr₃ TSM device-based crossbar array. Based on the 100 stable TSM devices, we characterize the device-to-device variation by measuring the HRS current of 100 samples. As shown in Supplementary Figure 4, the standard deviation of the HRS current was as low as 0.075. The narrow distribution of HRS current indicates high reproducibility of our device.

We added the relevant information to the manuscript and the Supporting Information as below.

In manuscript:

“To investigate the production yield of the FLBP-CsPbBr₃ TSM device-based crossbar array, we measured all the 400 devices in a 20×20 crossbar array. The *I*–*V* characteristics of the 400 memristor devices are provided in Supplementary Fig. 15-18, with 337 of them showing stable threshold switching characteristics and a production yield of 84.25%.”

In Supporting Information:

cc

Supplementary Figure 15. *I-V* characteristics of the first 100 FLBP-CsPbBr₃ TSM devices in a 20×20 crossbar array. All the *I-V* curves were obtained by sweeping the voltage in the sequence of 0 V → 3 V → 0 V.

Supplementary Figure 16. *I-V* characteristics of the second 100 FLBP-CsPbBr₃ TSM devices in a 20 × 20 crossbar array. All the *I-V* curves were obtained by sweeping the voltage in the sequence of 0 V → 3 V → 0 V.

Supplementary Figure 17. *I-V* characteristics of the third 100 FLBP-CsPbBr₃ TSM devices in a 20 × 20 crossbar array. All the *I-V* curves were obtained by sweeping the voltage in the sequence of 0 V → 3 V → 0 V.

Supplementary Figure 18. *I-V* characteristics of the fourth 100 FLBP-CsPbBr₃ TSM devices in a 20 × 20 crossbar array. All the *I-V* curves were obtained by sweeping the voltage in the sequence of 0 V → 3 V → 0 V.

Supplementary Figure 4. Spatial HRS current uniformity of the 100 stable FLBP-CsPbBr₃

TSM devices.

Based on the 100 stable TSM devices, we characterize the device-to-device variation by measuring the HRS current of 100 samples. As shown in Supplementary Fig. 4, the standard deviation of the HRS current was as low as 0.075. The narrow distribution of HRS current indicates high reproducibility of our device.”

2. In Figure 2a, the authors demonstrated a symmetrical bidirectional threshold switching characteristic. Why are the negative polarity regions in Figure 2a nearly the same with the positive polarity regions? Please add discussion in the manuscript.

ANS: In a representative sweep loop in Figure 2a (red curve for the positive voltage sweep and blue curve for the negative voltage sweep), the device transitioned to a LRS at 1.16 ± 0.24 V (\pm s.d.). In case active metals are used as electrodes, these metals may be doped into the dielectrics eventually under the synergistic effect of electric fields and thermal diffusion (*Adv. Funct. Mater.*, **2018**, 28, 1704862). Here, we use the active metal-silver as electrodes for our TSM device. The filament formation in the memristor is a ‘threshold’ event governed by the net Ag numbers in a given filamentary volume. The symmetrical bidirectional TS behaviors realized in our TSM device is attributed to the CF growth mode from the ITO electrode to Ag electrode. The mobility of Ag^+ is relatively high so that the generated Ag^+ from the Ag electrode could migrate rapidly to the ITO electrode and then, be reduced back to Ag atom. As the accumulation of Ag atoms, they constitute the nuclear of CFs and the CFs grow toward the Ag electrode to bridge the two electrodes eventually. When the CFs voluntarily rupture, a large quantity of Ag atoms remains near the ITO electrode side. When the polarity of bias is reversed, they can act as a new Ag electrode to accomplish a reverse TS behavior and the bidirectional TS is entirely realized in our TSM device. Thus, turn-on voltage at -1.19 ± 0.22 V (\pm s.d.) will be obtained during the following sweep in the negative bias, which indicating a stable symmetric bidirectional threshold switching behavior.

We add the relevant discussion for the symmetric bidirectional threshold switching behavior to the manuscript as below.

In the manuscript:

“Almost symmetric hysteresis loops were observed in both positive and negative scans, suggesting the unipolar TS characteristic which is significantly different from non-volatile electrochemical metallization memory (ECM). The active metal electrode material of Ag may be doped into the FLBP-CsPbBr₃ layer during the voltage sweeping process³⁶. As the filament formation in the memristor is a ‘threshold’ event governed by the net Ag numbers in a given filamentary volume. When the Ag CFs voluntarily rupture, the large amount of Ag atoms remains near the ITO electrode side. When the polarity of bias is reversed, they can act as a new Ag electrode to accomplish a reverse TS behavior and the bidirectional TS is entirely realized in our TSM device.”

3. Given the voltage response of the TSM neuron evoked by repetitive pulses, firing probability in TSM neuron extracted from experiments under different input pulse magnitude and pulse frequency are suggested to be provide.

ANS: According to the reviewer's suggestion, we extracted the measured firing events from experiments demonstrated in the Supplementary Fig. 28. The firing probability increases from 0.05, 0.075, 0.15 to 0.225 as the pulse magnitude increases from 1.5, 2, 2.5 to 3 V, respectively. Meanwhile, increasing stimulus frequency stimulus duration (from 3.3 Hz, 5 Hz, 10 Hz to 25 Hz) also improves the firing probability (from 0.083, 0.154, 0.158 to 0.222).

We added the firing probability in TSM neuron Supporting Information as below.

In the Supporting Information:

“The voltage response of TSM neuron evoked by repetitive pulses. a Experimental firing effect in TSM neuron as a function of the pulse magnitude. **The firing probability increases from 0.05, 0.075, 0.15 to 0.225 as the pulse magnitude increases from 1.5, 2, 2.5 to 3 V, respectively. b** Experimental firing effect in TSM neuron as a function of the pulse frequency. **The firing probability increases from 0.083, 0.154, 0.158 to 0.222 as the pulse magnitude increases from 3.3 Hz, 5 Hz, 10 Hz to 25 Hz, respectively.”**

4. The devices were fabricated on a flexible PDMS substrate. What is the relationship between bending angle and device properties?

ANS: To investigate the flexibility of the device, we performed the bending tests on the device with different bending angles. We added the relevant information to the Supporting Information as below.

In Supporting Information:

“The corresponding *I-V* characterization with varied bending angles are provided in Supplementary Fig. 5. The stable threshold switching performance as the device array is bending from 9° to 320° demonstrates the feasibility of the flexible applications.

Supplementary Figure 5. Typical I - V curves with respect to the different curvatures of the TSM. The optical images of memristor with different bending angles are shown in the upper panel.”

5. The authors claim that by comparing V_{spike} with V_F , the LGMD neuron can derive a directional decision from different situations. What is V_{spike} and V_F stand for? Please add discussion in the manuscript.

ANS: “ V_{spike} ” stands for “spiking frequency”. “ V_F ” stands for “firing frequency”, which is defined as 2.5 Hz in our manuscript. The defined value is based on the collision detecting spiking frequency in Fig. 6h. In order to give a better understanding, we corrected the “ V_{spike} ” as “ f_{spike} ”, and term “ V_F ” as 2.5 Hz in the manuscript as below.

In the manuscript:

“As shown in Fig. 6f, the spiking frequency (f_{spike}) strongly depends on the power of light. According to the previous results, the conductance of FLBP-CsPbBr₃ increases, peaks and decreases as the power of UV illumination increases (Fig. 6h). The f_{spike} follows the similar trend as the increase of UV irradiance which is analogue to the response of LGMD neuron to a looming object (Fig. 6g), and reaches to a 2.5 Hz (25 fired pulse within 10 s) to detect the subsequent collision. Whereas the influence of light wavelength on the f_{spike} is demonstrated in Supplementary Fig. 30. The f_{spike} decreases as the light wavelength shift from 365 nm to 520 nm.”

6. Given the large-scale deployment of neuromorphic computing in the future, it may be a good choice to take advantage of the mature silicon processing to be integrated with CMOS. The authors may state the implication of the current work for neuromorphic computing with considering CMOS compatibility.

ANS: As the reviewer mentioned, the CMOS compatible process with a low thermal budget is an advantage of the emerging two-terminal memristor device technology. In fact, the present experimental and simulation findings on the threshold switching phenomenon in the field of data storage memories opens the way for design of cross-point synapse array and implementing neural networks based on Si CMOS technology. (Prof. Jun Yao, *Nature Communications*, **2020**, 11:1861) Described by Prof. J. Joshua Yang, “A threshold switch based on such a phenomenon is of importance in electronic applications, which not only complements? the CMOS based circuits but also enables novel designs” (*Advanced Functional Materials*, **2018**, 28, 1704862), the efficient CMOS-emerging data storage materials integration (including 2D materials, biomaterials, organic materials, and semiconductor quantum dot materials) is still a tough challenge yet, which needs the joint effect of scientists in different fields.

Here, we use FLBP-CsPbBr₃ nanocomposition as the matrix materials of the functional threshold layer, which was prepared via the spin-coating method. Although this spin-coating

method prepared FLBP-CsPbBr₃ layer cannot be integrate with the CMOS technology, it is well-compatible with the flexible substrate for future flexible device. The unique nonlinear threshold switching behavior under optical or electrical excitation in our manuscript could lead to the development of performant new flexible neuromorphic devices to mimic signal processing of biological neurons.

7. The light intensity in Figure 6f,h is 0.72 mW, while Figure 6b use light intensity with the unit of mW/cm². Please unify the unit expression.

ANS: Per the reviewer's suggestion, we unify the unit expression.

REVIEWERS' COMMENTS

Reviewer #1 (Remarks to the Author):

After the first revision, in their manuscript titled 'Light-mediated threshold switching memristor for biomimetic Lobula giant movement detector neuron', Y. Wang et. al. have made significant changes and additions for a better read. The current version of the manuscript has better explanations and more characterization studies of the underlying physical mechanisms involved in the functioning of their prototype device. While the reviewer believes that the manuscript has the potential to be considered for publication in Nature Communications, there could be some more changes/additions that can be performed to enhance the reading quality of the manuscript further. These suggestions and comments are given below:

- 1) In the section "TSM-based biomimetic compound eye under the fixed stimuli", supplementary information 15-18 showing 337 working devices is a good addition showing the production yield. However, the axes of these plots are not clear since each one has 100 plots. Thus adding a plot showing the threshold voltage range of these devices similar to Fig. 2b will be helpful.
- 2) Fig. 5c shows the detected collision time before impact. Adding another set of bars to refer to the actual impact time will be helpful.
- 3) The caption given for Fig. 5c is "Detected collision time before impact by the FLBP-CsPbBr₃ TSM for different looming object speeds or the applied voltage pulse amplitude conditions". The reviewer struggle to understand where the effects of 'applied voltage pulse amplitude conditions' are shown in this figure. The same applies to Fig. 5b as well.
- 4) What is the information conveyed through Fig. 5b? Explaining the relationship of looming speed, incident angle and the increase in current value with looming speed, will be helpful.
- 5) The title of the manuscript is "Light-mediated threshold switching memristor for biomimetic Lobula giant movement detector neuron". Since there are implementations like flexible compound eye and real-time collision avoidance, giving a broader title may increase the readership.
- 6) As mentioned in the first review, authors are encouraged to reduce grammatical and spelling errors in their manuscript. For example, caption of Fig. 2a is not clear. This would provide a better read.

Reviewer #2 (Remarks to the Author):

Authors well revised their manuscript according to reviewers' comments. In my opinion now it is acceptable for publication in this journal.

Reviewer #3 (Remarks to the Author):

The paper is suggested for publication now

Point-by-Point Response to the Reviewers

Reviewer #1 (Remarks to the Author):

After the first revision, in their manuscript titled ‘Light-mediated threshold switching memristor for biomimetic Lobula giant movement detector neuron’, Y. Wang et. al. have made significant changes and additions for a better read. The current version of the manuscript has better explanations and more characterization studies of the underlying physical mechanisms involved in the functioning of their prototype device. While the reviewer believes that the manuscript has the potential to be considered for publication in Nature Communications, there could be some more changes/additions that can be performed to enhance the reading quality of the manuscript further. These suggestions and comments are given below:

1) In the section “TSM-based biomimetic compound eye under the fixed stimuli”, supplementary information 15-18 showing 337 working devices is a good addition showing the production yield. However, the axes of these plots are not clear since each one has 100 plots. Thus adding a plot showing the threshold voltage range of these devices similar to Fig. 2b will be helpful.

ANS: Thank you for reviewing our paper. We have revised the manuscript according to your suggestion as below.

In Supporting Information:

“

Supplementary Figure 19. Histogram of V_{th} of FLBP-CsPbBr₃ TSM devices in a 20×20 crossbar array under positive sweep with Gaussian fitting.”

2) Fig. 5c shows the detected collision time before impact. Adding another set of bars to refer to the actual impact time will be helpful.

ANS: As per the reviewer’s suggestion, we added the sets of bars to refer to the actual current and the impact time to Fig. 5b and 5c in the manuscript. We revised Fig.5 in the manuscript as below.

In the manuscript:

“

Fig. 5 Escape response of FLBP-CsPbBr₃ TSM. **a** Output current of the FLBP-CsPbBr₃ TSM in response to different looming object speed with different incident angles. The laser power ramp from 0.00 to 2.50 mW with different rate to represent the different looming object speed. **b** Inflection point values in the output current of the FLBP-CsPbBr₃ TSM for different looming object speeds or the different positioned device with specific incident angles. **c** Detected collision time before impact by the FLBP-CsPbBr₃ TSM for different looming object speeds or the different positioned device with specific incident angles.”

3) The caption given for Fig. 5c is “Detected collision time before impact by the FLBP-CsPbBr₃ TSM for different looming object speeds or the applied voltage pulse amplitude conditions”. The reviewer struggle to understand where the effects of ‘applied voltage pulse amplitude conditions’ are shown in this figure. The same applies to Fig. 5b as well.

ANS: We are sorry that we forgot to correct the caption for Fig. 5 in our last revision. According to the suggestion, we have revised the caption for Fig. 5 in the revised manuscript.

In the manuscript:

“**Fig. 5 Escape response of FLBP-CsPbBr₃ TSM. a** Output current of the FLBP-CsPbBr₃ TSM in response to different looming object speed with different incident angles. The laser power ramp from 0.00 to 2.50 mW with different rate to represent the different looming object speed. **b** Inflection point values in the output current of the FLBP-CsPbBr₃ TSM for different looming object speeds or the different positioned device with specific incident angles. **c** Detected collision time before impact by the FLBP-CsPbBr₃ TSM for different looming object speeds or the different positioned device with specific incident angles.”

4) What is the information conveyed through Fig. 5b? Explaining the relationship of looming speed, incident angle and the increase in current value with looming speed, will be helpful.

ANS: The non-monotonic response of current for all the measurements is demonstrated in Fig. 5a. Thus, we extract the inflection point values of all the TSM output current and mapped the relationship of looming speed, incident angle and the current value in Fig. 5b. As per the reviewer’s suggestion, we discussed the implication of the current value for the implementation with different looming speed, incident angle in the revised manuscript as below.

In the manuscript:

“Fig. 5b maps the inflection point values of TSM output current as per the data shown in Fig. 5a. As expected, monotonic trend of the current value was obtained as a function of the object looming speed for the given incident angles. The monotonical increment in the current value with the laser incident angles further demonstrate the most sensitive collision detection with the 90° incident angle. Therefore, the front position TSM on the flexible compound eye with the 90° incident angle offers optimal signal detection. In addition, 3D spatial distribution of the time-to-collision detection (τ_D) of our large FoV artificial vision system is demonstrated in Fig. 5c. The further optimizations of our flexible artificial vision system with incident light may potentially bring forth applications in the vision chip of autonomous robots for indicating collision.”

5) The title of the manuscript is “Light-mediated threshold switching memristor for biomimetic Lobula giant movement detector neuron”. Since there are implementations like flexible compound eye and real-time collision avoidance, giving a broader title may increase the readership.

ANS: Thank you for reviewing our paper. We appreciate your insightful comments on our research. We have revised the manuscript title according to your suggestions and believe that our revisions have improved the paper.

In the manuscript:

“**Memristor-based biomimetic compound eye for real-time collision detection**”

6) As mentioned in the first review, authors are encouraged to reduce grammatical and spelling errors in their manuscript. For example, caption of Fig. 2a is not clear. This would provide a better read.

ANS: Thanks for reviewer's suggestion. We have revised the mistakes in the manuscript.

Reviewer #2 (Remarks to the Author):

Authors well revised their manuscript according to reviewers' comments. In my opinion now it is acceptable for publication in this journal.

ANS: We thank the referee for approving the revised changes and endorsing our work.

Reviewer #3 (Remarks to the Author):

The paper is suggested for publication now

ANS: We thank the referee for the positive comments on our work and the revised changes.